# Biting Off More Than You Can Detect: Retrieval-Augmented Multimodal Experts for Short Video Hate Detection

## Abstract

Short Video Hate Detection (SVHD) is increasingly vital as hateful content — such as racial and gender-based discrimination — spreads rapidly across platforms like TikTok, YouTube Shorts, and Instagram Reels. Existing approaches face significant challenges: hate expressions continuously evolve, hateful signals are dispersed across multiple modalities (audio, text, and vision), and the contribution of each modality varies across different hate content. To address these issues, we introduce **MoRE** (**M**ixture **o**f **R**etrieval-augmented multimodal **E**xperts), a novel framework designed to enhance SVHD. MoRE employs specialized multimodal experts for each modality, leveraging their unique strengths to identify hateful content effectively. To ensure model's adaptability to rapidly evolving hate content, MoRE leverages contextual knowledge extracted from relevant instances retrieved by a powerful joint multimodal video retriever for each target short video. Moreover, a dynamic sample-sensitive integration network adaptively adjusts the importance of each modality on a per-sample basis, optimizing the detection process by prioritizing the most informative modalities for each instance. Our MoRE adopts an end-to-end training strategy that jointly optimizes both expert networks and the overall framework, resulting in nearly a twofold improvement in training efficiency, which in turn enhances its applicability to real-world scenarios. Extensive experiments on three benchmarks demonstrate that MoRE surpasses state-of-the-art baselines, achieving an average improvement of 6.91% in macro-F1 score across all datasets.

## Keywords

Short video hate detection, retrieval augmentation, mixture of multimodal experts.

## 1 Introduction

Media consumption trends have increasingly shifted toward short videos, particularly on platforms like TikTok, YouTube Shorts, and Instagram Reels [31, 65, 71]. As a dynamic and immersive communication medium, short video can significantly boost user engagement and capture a larger share of daily screen time [2, 4, 7, 31]. These videos seamlessly integrate diverse media modalities – such as audio, text, and vision – to convey information, exerting more substantial effects on mental health and social cohesion than content confined to a single modality.

However, this multimodal integration also enables the subtle and covert dissemination of hateful content[1], embedding harmful messages across various media forms. Hateful content in short videos often targets attributes like race, gender, or religion [11, 24, 44, 57, 67, 69] and can manifest through multiple modalities. Moreover, the prevalence of hateful content varies across modalities, with each contributing uniquely to its overall impact. The continual

---

[1]**Disclaimer**: *This paper contains discussions of violence and discriminatory content that may be disturbing to some readers.*

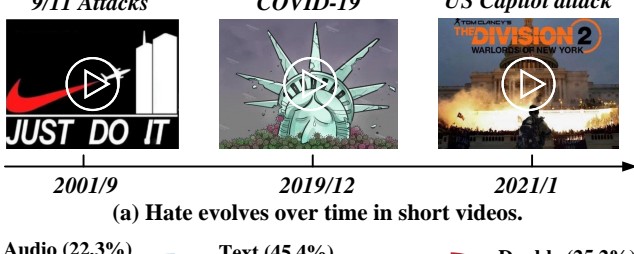

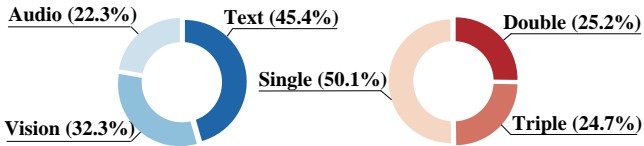

(a) Hate evolves over time in short videos.

(b) Distribution of hateful content across different modalities.

**Fig. 1: Illustration of motivation. (a): As new social events emerge, the expressions of hateful content undergo constant evolution. (b): Multimodal distribution of hateful content in the MHClip-B dataset [63]. The blue donut chart illustrates the distribution of hateful content across different modalities – audio, text, and vision. The red donut chart depicts the proportions of short videos that contain hateful content in one, two, or all three modalities.**

evolution of hateful content – driven by shifting social issues and advancements in tools for AI generated content (e.g., OpenAI's Sora [76]) – underscores the pressing need for highly effective methods to tackle the task of Short Video Hate Detection (SVHD).

Hateful content detection has been extensively studied in literature [1, 8–10, 14, 32, 40, 45, 48]. The majority of these works focus on text-based analyses [1, 14, 32, 48] within microblogging platforms such as Twitter and Facebook. With the increasing integration of images in social media posts and the advancements in image processing technologies, researchers have expanded their efforts to identify hateful elements in text-image posts and memes [8–10, 40, 45], utilizing pre-trained models and incorporating task-specific classification layers. However, despite the rapid rise in the popularity of short videos, research on hate detection in short videos remains very limited [13, 63]. Short videos encompass multiple modalities, which can subtly and covertly facilitate the dissemination of hateful content. In addition, the prevalence of hateful content varies across these modalities in short videos, which necessitates a dynamic and adaptable detection framework that can effectively identify hateful content across diverse modalities. Moreover, as hateful content is subject to continuous evolution, developing an effective and robust framework for SVHD entails addressing several significant challenges, which are summarized as follows:

**Challenge 1: Adapting to the Evolution of Hateful Content.** Hateful content continuously evolves in response to societal shifts, becoming more subtle and increasingly difficult to detect. Fig. 1(a) illustrates an evolution example through three short videos. Initially,

hate expressions employed imagery related to the 9/11 attacks to overtly criticize terrorism in the USA. Subsequently, during the COVID-19 pandemic, more nuanced and veiled content emerged, satirizing the response of American society. More recently, a combination of video game imagery and photos from the US Capitol attack has been utilized to critique American politics. This progression underscores the adaptive nature of hateful content over time. Consequently, it is imperative to develop detection frameworks that remain current and can generalize across increasingly evolving forms of hate in short videos.

**Challenge 2: Harnessing Multiple Modalities for Hateful Content Analysis.** Short videos encompass multimodal information such as audio, text, and visual content. Effectively utilizing data from different modalities for hate detection poses a significant challenge. The left side of Fig. 1(b) shows the modality-wise distribution of hateful content in the MHClip-B [63] dataset, highlighting that each modality contributes essential information for detecting hateful content, which can manifest in various forms. For instance, hate speech may be embedded in textual overlays, discriminatory lyrics may be presented in background music, and offensive gestures may appear in visual streams. Therefore, it is critical to develop a multimodal framework that can effectively integrate all modalities to detect various types of hateful content in short videos.

**Challenge 3: Managing Modality-Specific Influences in Hate Detection.** Not all modalities in short videos contribute equally to hate detection; each modality plays a distinct role. As shown in the right of Fig. 1(b), 75.3% short videos in the MHClip-B dataset contain hateful content presented in only one or two modalities. This distribution suggests that indiscriminately integrating all modalities could be counterproductive. The detection model may overemphasize noisy or redundant information, misleading the learning process and degrading detection performance. Thus, focusing on the most informative modalities and content is crucial for accurate detection. A more adaptive and selective multimodal fusion approach is needed to dynamically adjust each modality's contribution at the sample level, ensuring more precise hate detection.

To address these challenges, we propose a novel **M**ixture **o**f **R**etrieval-augmented multimodal **E**xperts (**MoRE**) framework. It introduces contextual knowledge-augmented multimodal experts designed to well adapt the dynamic and evolving hateful content and effectively harnesses data dispersed across multiple modalities in short videos for detection (i.e., Challenges 1 & 2). First, a basic expert is developed to focus on individual modalities, including audio, text, and vision. To adapt to the evolving nature of hateful content – mimicking human learning processes [27, 28] – our model retrieves relevant information to deepen its understanding on specific topics. The basic expert is subsequently augmented with contextual knowledge retrieved via a powerful joint multimodal video retriever, which integrates audio, textual, and visual modalities for fine-grained video-to-video retrieval. By leveraging contextual knowledge from the retrieved videos, the experts remain aware of the evolving expressions of hate, thereby enhancing their capability to generalize to emerging forms of hateful content. These contextual knowledge-augmented multimodal experts not only improves the adaptability of MoRE to new hate expressions but also ensures more accurate and comprehensive detection of hateful content across multiple modalities.

To address the varying significance of each modality in hate detection for different short videos (Challenge 3), MoRE incorporates a novel sample-sensitive integration network. This network includes a modality-mixture soft router which identifies the specific contributions of each modality's features to hate detection in each video, prioritizing those with the most significant impact for each video sample. Consequently, the network accurately determines the contributions of different modalities at the sample level, enhancing detection performance and providing interpretability regarding the roles of various modalities in hate detection for each short video.

Additionally, instead of a traditional two-stage training process [6, 70, 73], we introduce a unified and effective end-to-end training paradigm. This paradigm jointly optimizes both the experts and the overall framework, providing a scalable and applicable solution for SVHD. In summary, the key contributions of this work are as follows:

- **Contextual Knowledge-Augmented Multimodal Experts:** We design several multimodal experts to better adapt to the continuously evolving nature of hateful content in short videos and harness the multiple modalities in hate detection. By retrieving relevant instances through a powerful joint multimodal video retriever, the experts acquire contextual knowledge that deepens their understanding of specific topics, enabling them to keep pace with the evolving expressions of hate in short videos.

- **Sample-Sensitive Integration Network:** We propose a novel adaptive integration network that evaluates the varying contributions of different modalities within individual video samples to improve the performance of hate detection. This adaptive integration network dynamically adjusts the influence of each modality, prioritizing those with the most significant impact on detecting hateful content, thereby ensuring more precise and effective detection.

- **Unified End-to-End Training Paradigm:** We develop an effective end-to-end training paradigm that significantly enhances the model's scalability and applicability, making the model highly suitable for deployment in large-scale SVHD applications.

Extensive experiments on three real-world short video datasets demonstrate that MoRE outperforms state-of-the-art baselines. Notably, our model achieves an average improvement of 6.91% in macro-F1 score across all three datasets. Furthermore, our model surpasses three popular Large Vision-Language Models (LVLMs), highlighting its effectiveness and efficiency for SVHD, even when compared to large models trained on trillions of tokens and billions of parameters. The source codes and data required to reproduce our results are available at https://anonymous.4open.science/r/MoRE-SVHD and will be made public.

## 2 Related Work

Early studies primarily focused on identifying hate speech within text-based materials. Traditional machine learning approaches, such as Support Vector Machines and Naive Bayes classifiers [33, 66], have been commonly used for detection. With the rise of deep learning, more advanced methods have been developed for hate speech detection [1, 48]. Subsequently, multimodal hate detection, which analyzes both textual and visual information in posts and memes, has made significant progress [8–10, 40, 46]. For example,

Pro-Cap [8] leverages pre-trained models and prompting techniques to generate image captions that identify hateful content. Similarly, RGCL [46] learns hate-aware vision and language representations through a contrastive objective and retrieved examples. However, despite their effectiveness, these approaches are not directly applicable to hate detection in videos. Unlike text-image posts or memes, videos consist of multiple frames and incorporate various modalities, making it unclear which modality carries the hateful message, thereby highly increasing detection complexity.

Research on video-based hate detection remains limited. Recent advancements include the introduction of benchmark datasets such as HateMM [13] and MHClip [63]. Although baseline detection models were provided, they simply fused audio, text, and visual features equally for prediction. This simple design undermines their effectiveness in SVHD, as it overlooks the dynamic nature of hateful content and the varying significance of each modality in detecting hate across different short videos. In contrast, our proposed MoRE first retrieves the most relevant instances to construct the contextual knowledge-augmented multimodal experts that adapt to the evolving nature of hateful content. Then, a sample-sensitive integration network adaptively assigns weights to these experts at the sample level, further enhancing the prediction accuracy of MoRE in detecting hateful content in short videos.

Additional research related to the techniques used in MoRE, including multimodal retrieval and the Mixture of Experts (MoE), is reviewed in Appendix A.

## 3 Methodology

**Problem Statement.** Let $\mathcal{S} = \{S_1, \cdots, S_N\}$ denote the set of short videos on video platforms, where $N$ is the number of short videos. Each short video $S_i$ is characterized by its multimodal content, including audio, textual, and visual content, expressed as $S_i = \{s_i^a, s_i^t, s_i^v\}$. The objective of SVHD is to determine whether a given short video $S_i$ is **hateful** or **non-hateful** by considering all its modal contents $s_i^a$, $s_i^t$, and $s_i^v$.

**Feature Extraction.** The extracted features are summarized as follows: the audio features $\mathbf{x}_i^a \in \mathbb{R}^{l \times d_a}$, the visual features $\mathbf{x}_i^v \in \mathbb{R}^{m \times d_v}$, and the textual features $\mathbf{x}_i^t \in \mathbb{R}^{n \times d_t}$, where $l$ is the number of audio frames, $m$ is the number of key frames sampled from the video, and $n$ represents the number of word tokens. $d_a$, $d_t$, and $d_v$ are the feature dimensions for each modality. The detailed feature extraction process is provided in Appendix B.

Fig. 2 provides an overview of our proposed MoRE framework and illustrates the relationship among its core components. The following sections will delve into each component of MoRE, providing detailed explanations of their roles and interactions.

### 3.1 Joint Multimodal Video Retriever

To provide relevant instances to make our framework better adapt to the complex and evolving nature of hateful content, we design a novel joint multimodal video retriever, which simultaneously incorporates audio, textual, and visual features to perform video-to-video retrieval, moving beyond the limitations of unimodal retrieval methods that rely on a single modality. By jointly considering all modalities, our strategy enables the retrieval of instances associated with the target video from multiple perspectives, leading to significantly improved retrieval precision.

*3.1.1 Memory Bank Construction.* To store high-quality semantic information as prior knowledge, we define the memory bank $\mathcal{B}$, which encodes audio, textual, and visual content using a collection of (audio, text, vision) triples. The memory bank $\mathcal{B}$ is typically extensive and encompasses a wide array of concepts, yet only a small subset is relevant to a given video. We focus on these most pertinent instances for each short video.

*3.1.2 Query Construction.* To fully capture the unique characteristics of each modality, we first encode the audio, textual, and visual features independently. Specifically, for each short video $S_i$, we first extract its audio transcription using Whisper [53], a pre-trained automatic speech recognition model. The transcription is then processed by a pre-trained BERT [16] model to generate the audio query vector $\mathbf{r}_i^a \in \mathbb{R}^{d_a}$. For textual retrieval, we use the BERT model to extract semantic features from the concatenated title and description of $S_i$, resulting in the textual query vector $\mathbf{r}_i^t \in \mathbb{R}^{d_t}$. Finally, for visual retrieval, we input the key frames of $S_i$ into a pre-trained Vision Transformer (ViT) [19] and average the frame representations to generate the visual query vector $\mathbf{r}_i^v \in \mathbb{R}^{d_v}$.

*3.1.3 Weighted Similarity-based Multimodal Retrieval.* To effectively and comprehensively capture the relevance across audio, textual, and visual modalities, we propose a weighted similarity-based multimodal retrieval strategy. Specifically, given a short video $S_i$, we compute a weighted similarity score that integrates the similarities from audio, textual, and visual queries. The similarity score between two videos $S_i$ and $S_j$ is computed as:

$$\text{Score} = w_a \cdot \text{sim}(\mathbf{r}_i^a, \mathbf{r}_j^a) + w_v \cdot \text{sim}(\mathbf{r}_i^v, \mathbf{r}_j^v) + w_t \cdot \text{sim}(\mathbf{r}_i^t, \mathbf{r}_j^t), \quad (1)$$

where $w_a$, $w_v$, and $w_t$ are the weights assigned to the similarity of each modality; $\mathbf{r}_i^a$, $\mathbf{r}_i^v$, and $\mathbf{r}_i^t$ represent the audio, visual, and textual query vectors for video $S_i$, respectively. The similarity function $\text{sim}(\mathbf{x}_1, \mathbf{x}_2)$ is defined as:

$$\text{sim}(\mathbf{x}_1, \mathbf{x}_2) = \frac{\mathbf{x}_1^\top \mathbf{x}_2}{\|\mathbf{x}_1\| \cdot \|\mathbf{x}_2\|}. \quad (2)$$

After calculating the similarity scores between $S_i$ and each short video stored in $\mathcal{B}$, the top-$K$ most similar hateful videos $S_i^r = \{S_i^{r_j}\}_{j=1}^K$ and the top-$L$ most similar non-hateful videos $\bar{S}_i^r = \{\bar{S}_i^{r_j}\}_{j=1}^L$ are selected as retrieval results. These retrieved instances provide contextual knowledge, empowering modality experts to more effectively address the evolving nature of hateful content in short videos, which will be discussed in the next section.

### 3.2 Contextual Knowledge-Augmented Multimodal Experts

In the context of MoE, the experts represent neural networks designed to tackle particular types of tasks or data patterns. To begin with, we propose the multimodal experts networks, where each expert network is assigned to process a specific modality. Specifically, following the previous works [6, 70, 75], we simply define three modality experts, where each expert network adopts a feed-forward network (FFN) structure to capture modality-specific features,

$$\mathbf{E}_i^m = \text{FFN}(\mathbf{x}_i^m), \quad (3)$$

$$\text{FFN}(\mathbf{x}) = (\text{ReLU}(\mathbf{x}\mathbf{W}_1 + b_1))\,\mathbf{W}_2 + b_2, \quad (4)$$

**Fig. 2: Overall framework of MoRE. (1): The joint multimodal video retriever identifies similar instances by considering all the modalities. (2): The contextual knowledge-augmented multimodal experts are designed to utilize retrieved information from (1) to adapt to evolving hate expressions, while leveraging all the modalities for accurate detection. (3): The sample-sensitive integration network provides a flexible mixture to allocate weights to each expert in (2). "H": Hateful, "N": Non-hateful.**

where $m \in \{a, t, v\}$ denotes the type of modality, $\mathbf{E}_i^m \in \mathbb{R}^{s \times d}$ is the representation of the modality expert for the short video $S_i$, $s$ is the sequence length, and $d$ is the feature dimension.

A **significant limitation** of the vanilla experts in prior works lies in their inability to adapt to the evolving nature of hateful content. To address this, we propose the Bipolar Hateful Attention Network (BHAN), which equips contextual knowledge from relevant videos to the vanilla experts to make them "up-to-date". Inspired by contrastive learning, BHAN utilizes hateful and non-hateful instances retrieved from the memory bank $\mathcal{B}$, equipping experts with contextual knowledge from both types of content. By leveraging these contrasting examples, BHAN empowers the experts to stay responsive to the ongoing shifts in hateful behavior and to capture the subtle distinctions between hateful and non-hateful content.

Specifically, for each modality expert, we first feed the retrieved hateful modality features $\mathbf{x}_i^{m,r} = \{\mathbf{x}_i^{m,r_j}\}_{j=1}^K$ and non-hateful modality features $\bar{\mathbf{x}}_i^{m,r} = \{\bar{\mathbf{x}}_i^{m,r_j}\}_{j=1}^L$ into the FFN to obtain the embeddings $\mathbf{E}_i^{m,r}$ and $\bar{\mathbf{E}}_i^{m,r}$, where $m \in \{a, t, v\}$. To equip the modality expert representation $\mathbf{E}_i^m$ with bipolar contextual knowledge, we introduce two attention mechanisms: $\text{Att}_{\text{Hat}}$ for hateful and $\text{Att}_{\text{Non}}$ for non-hateful attention. This process can be formalized as:

$$\tilde{\mathbf{E}}_i^{m,c} = \text{Att}_{\text{Hat}}(\mathbf{E}_i^m, \mathbf{E}_i^{m,r}, \mathbf{E}_i^{m,r}) + \text{Att}_{\text{Non}}(\mathbf{E}_i^m, \bar{\mathbf{E}}_i^{m,r}, \bar{\mathbf{E}}_i^{m,r}) + \mathbf{E}_i^m, \quad (5)$$

with the attention mechanisms $\text{Att}_{\text{Hat}}$ and $\text{Att}_{\text{Non}}$ defined as:

$$\text{Att}_{\text{Hat}}(\mathbf{Q}, \mathbf{K}, \mathbf{V}) = \alpha \cdot \text{Softmax}\left(\frac{\mathbf{Q}\mathbf{K}^T}{\sqrt{d}}\right)\mathbf{V}, \quad (6)$$

$$\text{Att}_{\text{Non}}(\mathbf{Q}, \mathbf{K}, \mathbf{V}) = (1 - \alpha) \cdot \text{Softmax}\left(\frac{\mathbf{Q}\mathbf{K}^T}{\sqrt{d}}\right)\mathbf{V}, \quad (7)$$

where $\alpha$ denotes the balance between the hateful and non-hateful attention contributions. We then apply an attentive pooling strategy [60] to $\tilde{\mathbf{E}}_i^{m,c} \in \mathbb{R}^{s \times d}$ across the sequence dimension to obtain

the representation of the contextual knowledge-augmented multimodal experts $\mathbf{E}_i^{m,c} \in \mathbb{R}^d$ for the short video $S_i$.

### 3.3 Sample-Sensitive Integration Network

The considerable variability in modality characteristics across different short videos, along with the fact that the importance of each modality varies significantly for detecting hateful content in different videos, jointly pose a major challenge for traditional modal fusion techniques in SVHD. These methods typically apply equal weighting to all modalities, disregarding the variation in modal contributions across different samples during prediction. To address this, we propose a sample-sensitive integration network that adaptively assigns weights to each modality expert based on the unique characteristics of input video samples, prioritizing the most influential modalities for detecting hateful content in each video.

Specifically, we first employ a non-parametric strategy by applying average pooling to the original representations of each modality, resulting in comprehensive representations for the three modalities: $\tilde{\mathbf{x}}_i^a \in \mathbb{R}^{d_a}$, $\tilde{\mathbf{x}}_i^t \in \mathbb{R}^{d_t}$, and $\tilde{\mathbf{x}}_i^v \in \mathbb{R}^{d_v}$. Subsequently, we align the modal dimensions to a uniform size and design a Modality-mixture Soft Router (MSR) — i.e., a two-layer MLP — to generate dynamic weights for the fusion of the multimodal experts $\mathbf{E}_i^{a,c}$, $\mathbf{E}_i^{t,c}$, and $\mathbf{E}_i^{v,c}$ at the sample-level. This process yields the final representation for the short video $S_i$, which can be expressed as:

$$\tilde{\mathcal{W}}_i = [\tilde{w}_i^a, \tilde{w}_i^t, \tilde{w}_i^v] = \text{MSR}([\Psi_a(\tilde{\mathbf{x}}_i^a), \Psi_t(\tilde{\mathbf{x}}_i^t), \Psi_v(\tilde{\mathbf{x}}_i^v)]), \quad (8)$$

$$w_i^m = \text{Softmax}(\tilde{w}_i^m) = \frac{e^{\tilde{w}_i^m}}{\sum_{j \in \{a,t,v\}} e^{\tilde{w}_i^j}}, \quad (9)$$

$$\mathbf{E}_i = \sum_{m \in \{a,t,v\}} w_i^m \cdot \mathbf{E}_i^{m,c}, \quad (10)$$

where [,] is the concatenation operation, $\Psi_a(\cdot)$, $\Psi_t(\cdot)$ and $\Psi_v(\cdot)$ denote the linear mapping functions, $w_i^m$ represents the weight

assigned to each modality expert for short video $S_i$, and $\mathbf{E}_i \in \mathbb{R}^d$ is the final representation for prediction. $\mathbf{E}_i$ is then fed into a predictor (i.e., a two-layer MLP with an activation function) to generate the classification result for short video $S_i$: $\hat{y}_i = \text{Predictor}(\mathbf{E}_i)$.

## 3.4 End-to-End Training

Previous MoE-based approaches [6, 70, 73] commonly employ a two-stage training paradigm. Each expert network is trained independently in the first stage, and in the second stage, these experts are integrated with a router network for joint optimization. While this approach allows for comprehensive expert training, it introduces considerable computational overhead by separate optimization phases, limiting its efficiency in real-world applications.

In contrast, we propose a more efficient and practically applicable **end-to-end training paradigm**, where the expert networks and the overall framework are optimized jointly, leading to greater computational efficiency. Specifically, we define the classification outputs from each modality expert as $\hat{y}_i^a$, $\hat{y}_i^t$, and $\hat{y}_i^v$. The joint training process is formulated as:

$$L_{\text{joi}} = \min\{1 - f_{\text{epo}}, 1 - \delta\} \cdot L_{\text{exp}} + \max\{f_{\text{epo}}, \delta\} \cdot L_{\text{ovl}}, \quad (11)$$

$$L_{\text{exp}} = \sum_{m \in \{a,t,v\}} L_{\text{BCE}}(\hat{y}_i^m, y_i), \quad (12)$$

$$L_{\text{ovl}} = L_{\text{BCE}}(\hat{y}_i, y_i), \quad (13)$$

where $L_{\text{exp}}$ represents the training loss for the expert networks and $L_{\text{ovl}}$ denotes the loss for overall framework. $\delta$ represents a small positive constant (non-zero), used to ensure stability during training. $L_{\text{BCE}}$ is the binary cross-entropy loss. The smoothly varying weight function $f_{\text{epo}} = (\text{epoch}_{\text{current}}/\text{epoch}_{\text{total}})^2$ modulates the focus of the loss during training, placing greater emphasis on modality expert training during the early stages and gradually shifting toward optimizing the entire network in the later stages. The details of the computational complexity analysis, the training strategy and algorithm, as well as the mathematical proof of the effectiveness of MoRE are provided in Appendix C-F.

## 4 Experiments

In this section, we conduct extensive experiments to verify the efficacy of MoRE. Initially, we provide an overview of the datasets, the baselines, and the metrics used, with details regarding datasets, baselines, and implementation available in Appendix G. Additional experiments are presented in Appendix H.

**Datasets**. To evaluate the efficacy of the proposed MoRE, we conduct comprehensive experiments on three real-world short video datasets, including HateMM [13], MultiHateClip-Youtube (MHClip-Y) and MultiHateClip-Bilibili (MHClip-B) [63]. Detailed dataset statistics are presented in Table 1, with splits aligned with the original paper. $\mathbf{A}$, $\mathbf{T}$, and $\mathbf{V}$ denote the dimension of audio, textual, and visual features, respectively.

**Baselines**. We compare MoRE with 13 competitive baselines, which can be categorized into three distinct groups: (1) *Unimodal hate detection methods*, which utilize a single modality for hate detection, including BERT [16], ViViT [3], MFCC [15], and SharedCon [1]. (2) *Multimodal hate detection methods*, which incorporate all available modalities within the short video to enhance the prediction accuracy, including Pro-Cap [8], HTMM [13], RGCL [46], MHCL [63],

Mod-HATE [10] and ExplainHM [40]. (3) *Large Vision-Language Model (LVLM)-based methods*, which leverage task-agnostic multimodal pre-training and demonstrate superior performance in visual question answering and video captioning, including the recently released MiniCPM-V [72], LLaVA-OV [36], and Qwen2-VL [64].

**Metrics**. Following prior works [13, 63], we adopt four metrics in SVHD to comprehensively evaluate the model's performance: classification Accuracy (**ACC**), Macro-F1 score (**M-F1**), Macro Precision (**M-P**) and Macro Recall (**M-R**).

**Table 1: Statistics of three short video datasets.**

| Dataset | # Total | # Train | # Val | # Test | A | T | V |
|---------|---------|---------|-------|--------|-----|-----|-----|
| HateMM | 1,083 | 757 | 109 | 217 | 128 | 768 | 768 |
| MHClip-Y | 1,000 | 700 | 100 | 200 | 128 | 768 | 768 |
| MHClip-B | 1,000 | 699 | 101 | 200 | 128 | 768 | 768 |

## 4.1 Overall Performance

To verify the superiority of our MoRE, we compare it with 13 competitive baselines on three datasets and the results are reported in Table 2. From these results, we have the following observations:

**(O1)**: **Multimodal hate detection methods generally outperform the unimodal methods.** Unimodal methods only leverage single modality for prediction, which is prone to missing essential information and overlooking hateful content manifesting in other modalities, leading to weak performance. Multimodal detection methods leverage features from all the modalities to improve the precision of prediction. Moreover, MoRE performs best among multimodal methods, as these multimodal baselines typically overlook the evolving nature of hateful content, which requires the model to remain current. Furthermore, these methods often adopt a vanilla fusion strategy that treats modalities equally in modal fusion, overlooking the varying importance of each modality across different instances in SVHD, which requires a more flexible fusion approach.

**(O2)**: **LVLM-based methods exhibit strong performance in SVHD.** LVLM-based methods have recently gained prominence due to their impressive performance across a wide range of multimodal tasks. These methods leverage the latest advanced LVLMs, whose effectiveness largely stems from extensive pre-training on large-scale vision-language corpora, enabling them to generalize well across many multimodal downstream tasks. Despite their strong capability in detecting hate in short videos, MoRE outperforms these models due to the lack of task-specific adaptation in LVLMs required for SVHD.

**(O3)**: **MoRE outperforms all strong baseline models across three datasets.** Notably, MoRE achieves average improvements of 5.27% in ACC and 6.91% in M-F1 across all three datasets. To further validate MoRE's superiority, we compute the statistical differences between MoRE and the best-performing baseline by retraining both models five times. The resulting $p$-values, all below 0.05, confirm that MoRE's improvement over the baseline is statistically significant. These performance gains demonstrate the effectiveness of incorporating expressive contextual knowledge from retrieved instances, which enables the experts to adapt to the evolving nature of hateful content and enhances their discriminative power. Moreover, the sample-sensitive integration network dynamically allocates contribution for each expert based on the characteristics of each video sample, leading to further improvements in SVHD.

**Table 2: Experimental results of the competitive baseline models and the proposed MoRE on the HateMM, MHClip-Y and MHClip-B datasets. ACC: Accuracy, M-F1: Macro-F1 score, M-P: Macro Precision, M-R: Macro Recall. The best results are in red bold, while the second results are in black bold. Higher values of ACC, M-F1, M-P, and M-R indicate better performance.**

| Method | HateMM | | | | MHClip-Y | | | | MHClip-B | | | |
|---|---|---|---|---|---|---|---|---|---|---|---|---|
| | ACC | M-F1 | M-P | M-R | ACC | M-F1 | M-P | M-R | ACC | M-F1 | M-P | M-R |
| BERT | 0.6912 | 0.6368 | 0.7008 | 0.6396 | 0.6547 | 0.4909 | 0.5522 | 0.5220 | 0.7251 | 0.6771 | 0.6839 | 0.6279 |
| ViViT | 0.6820 | 0.6670 | 0.6682 | 0.6661 | 0.6705 | 0.6143 | 0.6215 | 0.6111 | 0.7099 | 0.6610 | 0.6661 | 0.6575 |
| MFCC | 0.6543 | 0.6031 | 0.6410 | 0.6069 | 0.6650 | 0.4715 | 0.5877 | 0.5222 | 0.6307 | 0.5250 | 0.5410 | 0.5304 |
| SharedCon | 0.6956 | 0.6857 | 0.6872 | 0.6846 | 0.6850 | 0.6478 | 0.6484 | 0.6473 | 0.7250 | 0.7089 | 0.7325 | 0.7069 |
| Pro-Cap | 0.6451 | 0.6326 | 0.6335 | 0.6321 | 0.7006 | 0.6633 | 0.6633 | 0.6633 | 0.7250 | 0.6677 | 0.6606 | 0.6832 |
| HTMM | 0.7603 | 0.7278 | 0.7794 | 0.7201 | 0.7153 | 0.6319 | 0.6830 | 0.6264 | 0.7102 | 0.6183 | 0.6654 | 0.6136 |
| RGCL | 0.7558 | 0.7355 | 0.7296 | 0.7524 | 0.7133 | 0.6322 | 0.6264 | 0.6728 | 0.7250 | 0.7103 | 0.7365 | 0.7096 |
| MHCL | **0.7741** | **0.7654** | 0.7649 | 0.7659 | 0.7103 | 0.6547 | 0.6722 | 0.6486 | **0.7650** | 0.7311 | 0.7320 | **0.7302** |
| Mod-HATE | 0.6866 | 0.6536 | 0.6510 | 0.6760 | 0.6774 | 0.6388 | 0.6376 | 0.6702 | 0.6445 | 0.5107 | 0.5233 | 0.5046 |
| ExplainHM | 0.7315 | 0.6888 | 0.6822 | 0.7013 | 0.7250 | 0.6700 | 0.6890 | 0.6737 | 0.7500 | 0.7295 | 0.7318 | 0.7249 |
| MiniCPM-V | 0.7235 | 0.7228 | 0.7781 | 0.7635 | 0.6910 | 0.6742 | 0.6929 | **0.6740** | 0.7157 | 0.7015 | 0.7359 | 0.7044 |
| LLaVA-OV | 0.7558 | 0.7557 | 0.7790 | **0.7828** | **0.7350** | **0.6766** | **0.7045** | 0.6674 | 0.7521 | 0.7078 | 0.7143 | 0.7031 |
| Qwen2-VL | 0.7373 | 0.7371 | **0.7805** | 0.7732 | 0.7050 | 0.6677 | 0.6684 | 0.6671 | 0.7601 | **0.7326** | **0.7385** | 0.7285 |
| **MoRE** | **0.8341** | **0.8235** | **0.8178** | **0.8334** | **0.7750** | **0.7519** | **0.7567** | **0.7482** | **0.7850** | **0.7475** | **0.7568** | **0.7410** |
| Improv. | 7.75%↑ | 7.59%↑ | 4.78%↑ | 6.46%↑ | 5.44%↑ | 11.13%↑ | 7.41%↑ | 11.01%↑ | 2.61%↑ | 2.03%↑ | 2.48 %↑ | 1.48%↑ |
| $p$-val. | $9.72e^{-3}$ | $8.52e^{-3}$ | $7.44e^{-3}$ | $7.51e^{-3}$ | $9.91e^{-4}$ | $3.07e^{-4}$ | $1.47e^{-3}$ | $3.67e^{-4}$ | $2.29e^{-4}$ | $1.68e^{-3}$ | $2.62e^{-4}$ | $3.61e^{-3}$ |

**Table 3: Ablation study on core components within MoRE. The best results are in black bold.**

| Variant | HateMM | | MHClip-Y | | MHClip-B | |
|---|---|---|---|---|---|---|
| | ACC | M-F1 | ACC | M-F1 | ACC | M-F1 |
| Uni Retriever | 0.7972 | 0.7744 | 0.7402 | 0.6810 | 0.7790 | 0.7303 |
| w/o Retriever | 0.7557 | 0.7355 | 0.6950 | 0.6637 | 0.7150 | 0.6836 |
| BHAN-Att$_{Hat}$ | 0.8018 | 0.7887 | 0.7610 | 0.6881 | 0.7550 | 0.7009 |
| BHAN-Att$_{Non}$ | 0.8110 | 0.7985 | 0.7550 | 0.7240 | 0.7750 | 0.7358 |
| w/o BHAN | 0.7880 | 0.7723 | 0.7315 | 0.7120 | 0.7001 | 0.6581 |
| w/o Router | 0.7882 | 0.7734 | 0.7302 | 0.6815 | 0.7211 | 0.6902 |
| **MoRE** | **0.8341** | **0.8235** | **0.7750** | **0.7519** | **0.7850** | **0.7475** |

## 4.2 Ablation Study

To further understand the roles of core components and multimodal experts in our proposed MoRE framework, comprehensive ablation studies are conducted.

*4.2.1 Ablation Study on Core Components.* We conduct an ablation study to analyze the role of each core component within MoRE, and the results are summarized in Table 3.

**Effect of joint multimodal video retriever.** To validate the efficacy of the joint multimodal video retriever, we designed two variant models: (1) **Uni Retriever**: replacing multimodal joint video retriever with an unimodal retriever, performing text-to-text retrieval, and (2) **w/o Retriever**: removing the retriever entirely by using random samples to replace the retrieved instances. The results demonstrate that unimodal retrieval, limited to a single modality, fails to capture the most relevant instances, leading to suboptimal performance. Furthermore, completely removing the retrieval process causes a substantial drop in performance, highlighting the crucial role of high-quality retrieved instances. In contrast, our multimodal joint video retriever, which incorporates information from

all three modalities, consistently improves the retrieval quality and strengthens the overall framework performance.

**Effect of contextual knowledge-augmented multimodal experts.** To analyze the impact of contextual knowledge equipped to the modality experts, we design three variant models: (1) **BHAN-Att$_{Hat}$**: removing the non-hateful attention from the BHAN, (2) **BHAN-Att$_{Non}$**: removing the hateful attention from the BHAN, and (3) **w/o BHAN**: removing the BHAN entirely. The removal of each type of attention results in a notable performance drop, highlighting the importance of integrating contextual knowledge from both hateful and non-hateful relevant instances. Moreover, eliminating the entire BHAN leads to a substantial performance decline, underscoring the critical role of equipping modality experts with contextual insights, which facilitates the experts to adapt the ever-changing hate and improve their ability to distinguish the subtle difference between content of hate and non-hate.

**Effect of sample-sensitive integration network.** We evaluate the impact of the sample-sensitive integration network by designing the variant model: **w/o Router**: replacing the router network with a simple sum-based fusion method. The results indicate that equally fusing the modalities fails to accurately detect hate in short videos. In fact, the hateful content may manifest in different modalities, which necessitates a flexible fusion approach, the sample-sensitive integration network, to dynamically assign the modal contribution for each short video instance.

*4.2.2 Ablation Study on Multimodal Experts.* The second ablation study evaluates the contribution of each modality expert in detecting hateful content. It employs various combinations of modality experts in MoRE, with the results presented in Table 4. Based on these results, we have the following observations:

**(O1): Different modal experts have significantly different impacts.** Across all three datasets, we observe significant variability

**Table 4: Ablation study on multimodal experts within MoRE. The best results are in black bold. A: Audio expert, T: Textual expert, V: Visual expert.**

| Expert(s) | HateMM | | MHClip-Y | | MHClip-B | |
|---|---|---|---|---|---|---|
| | ACC | M-F1 | ACC | M-F1 | ACC | M-F1 |
| { A } | 0.6451 | 0.5826 | 0.6521 | 0.5132 | 0.6497 | 0.4531 |
| { T } | 0.7188 | 0.6972 | 0.7350 | 0.6765 | 0.7201 | 0.6880 |
| { V } | 0.6866 | 0.6415 | 0.7002 | 0.5888 | 0.7150 | 0.6557 |
| { A, T } | 0.7281 | 0.7004 | 0.7250 | 0.6491 | 0.7305 | 0.6614 |
| { A, V } | 0.7373 | 0.6935 | 0.6651 | 0.4132 | 0.6850 | 0.5771 |
| { T, V } | 0.8110 | 0.7954 | 0.7402 | 0.6739 | 0.7502 | 0.7252 |
| **MoRE** | **0.8341** | **0.8235** | **0.7750** | **0.7519** | **0.7850** | **0.7475** |

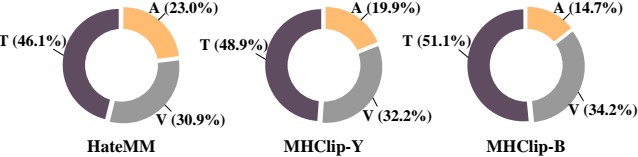

**Fig. 3: Visualization of modality experts contribution allocation of MoRE across all three datasets. A: Audio expert, T: Textual expert, V: Visual expert.**

in the impact of each expert. The textual expert consistently plays a more crucial role in SVHD compared to the visual and audio experts, with the audio expert contributing the least. This observation also aligns with the distribution of hateful content across each modality in the dataset, as exemplified by the MHClip-B dataset shown in the blue donut chart of Fig. 1(b).

**(O2): Effectively combining all experts brings better performance.** We observe that combining multiple experts consistently improves performance compared to using a single expert. In particular, combining textual and visual experts outperforms combining audio with either modality expert, reinforcing the relative weakness of the audio expert. Notably, our proposed MoRE effectively integrates all three experts through the sample-sensitive integration network to achieve optimal performance in SVHD.

To provide further insight into how MoRE leverages three modality experts, we present the average weight assigned by the router network in MoRE to each expert across different datasets in Fig. 3. The router consistently assigns the highest weight to the textual expert, followed by the visual expert, with the audio expert receiving the lowest weight. It demonstrates that the router can effectively adapt to the strengths of each expert, thereby providing an intuitive explanation for the observed improvements in MoRE performance.

## 4.3 Hyper-Parameter Analysis

This experiment presents the sensitivity analysis of MoRE's hyperparameters, specifically the number of retrieved videos $K$ and $L$, on the HateMM and MHClip-Y datasets. Fig. 4 demonstrates that the performance of MoRE is improved by adding contextual knowledge retrieved from hateful and non-hateful videos to the modality expert. However, injecting a large number of instances to the expert may result in a performance decline due to the noise (i.e., the irrelevant videos). To achieve the optimal performance, the number

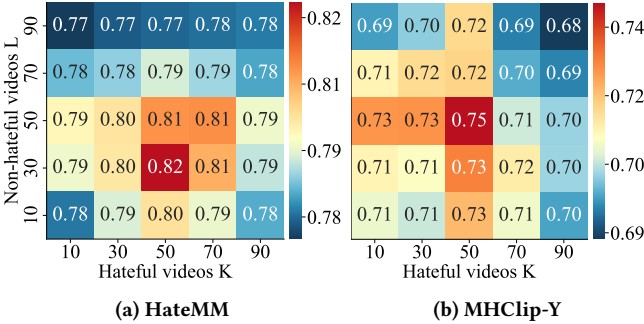

**Fig. 4: Sensitivity analysis of $K$ and $L$ on the HateMM and MHClip-Y datasets based on M-F1.**

**Table 5: Presentation of the retrieval quality. H: Hateful, N: Non-hateful. "V / A / T" refers to the cosine similarity scores between the target video and the retrieved videos across visual, audio, and textual modalities.**

| | Target: H | Top-1: H | Top-1: N |
|---|---|---|---|
| **Vision** | | | |
| **Audio** | I'm a prostitute. I don't charge body for sex. I give man a way for free... | That is, I look like a prostitute and I am charging the man for sex... | In Greek legend, Phryne, famous prostitute, the god give the body... |
| **Text** | I give a way for free; I am a lady; Mom called me a prostitute... | I am a prostitute; I am a lady of the evening dropped pants... | In Ancient Greek Prostitute; famous prostitute avoid showing... |
| **V / A / T** | N/A | 0.75 / 0.93 / 0.90 | 0.81 / 0.89 / 0.87 |

of retrieved hateful instances is set to $K = 50$ for both datasets. For the number of retrieved non-hateful videos, $L$ is set to 30 for the HateMM dataset and 50 for the MHClip-Y dataset. Further analysis of other parameters is provided in Appendix H.1.

## 4.4 Retrieval Quality Presentation

To validate the effectiveness of the proposed joint multimodal video retriever, we randomly select a hateful video from the test set of the MHClip-Y dataset with the retrieved results. As illustrated in Table 5, we observe that both hateful (H) and non-hateful (N) instances exhibit similar backgrounds and subjects, specifically featuring a woman speaking, which closely aligns with the target video's visual information. Furthermore, the texts and audio transcriptions of the retrieved instances show significant content overlap with the target video, including keywords such as "prostitute", "body", and "sex". This observation underscores the efficacy of our multimodal retrieval strategy, which seamlessly integrates all three

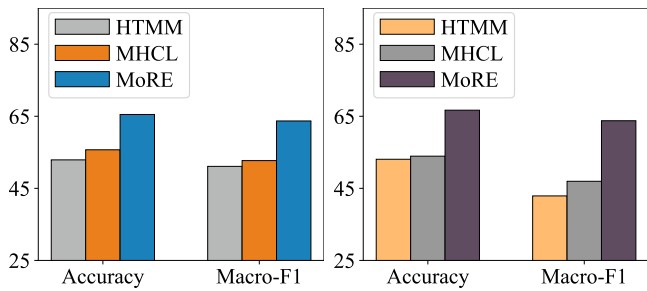

(a) From HateMM to MHClip-Y.   (b) From MHClip-Y to HateMM.

Fig. 5: Generalizability between the baselines: HTMM, MHCL, and our MoRE on the HateMM and MHClip-Y datasets

modalities to retrieve the most relevant instances. Notably, the content within the text and audio transcriptions in retrieved instances shares overlapping keywords with the target video, such as "prostitute". However, the hateful instance employs harmful and offensive language (e.g., "charging", "sex", "lady"), in stark contrast to the non-hateful instance, which engages in neutral discourse, such as the historical story of the prostitute in "Ancient Greece". Consequently, by effectively learning the nuanced distinctions between hateful and non-hateful instances, the modality experts are endowed with enhanced discriminative capabilities. Additional presentations of retrieved instances are presented in Appendix H.5.

## 4.5 Model Generalizability

To investigate the generalizability of MoRE and two competitive baseline models, particularly their ability to adapt to the new form of hateful content, we conduct experiments where the models are trained on one dataset and tested on the other. The HateMM and MHClip-Y datasets are selected due to the significant differences in their video content, stemming from their origins on entirely distinct online platforms. In these experiments, the memory bank $\mathcal{M}$ of MoRE is constructed using the training set of the target dataset. Initially, the models are trained on HateMM and tested on MHClip-Y, and subsequently, this setup is reversed to train on MHClip-Y and test on HateMM. The results are presented in Fig. 5.

Both baseline models demonstrate extremely weak performance when confronted with previously unseen hateful content, primarily due to their lack of design for handling generalization. In contrast, the proposed MoRE exhibits remarkable adaptability to these new forms of hate, as it leverages contextual knowledge from retrieved instances in the target dataset to enable the multimodal experts to effectively detect "unencountered" hateful content. These findings further confirm the superiority of MoRE in adapting to the evolving nature of hate in short videos and its ability to meet real-world demands by training on one platform and generalizing across multiple platforms.

## 4.6 Case Study: Model Explainability

In this section, we explore the explainability of MoRE by conducting a case study on two randomly selected hateful short videos from the test set of the MHClip-Y dataset. This case study aims to elucidate how MoRE adaptively assigns weights to multimodal experts to achieve accurate predictions for different video samples.

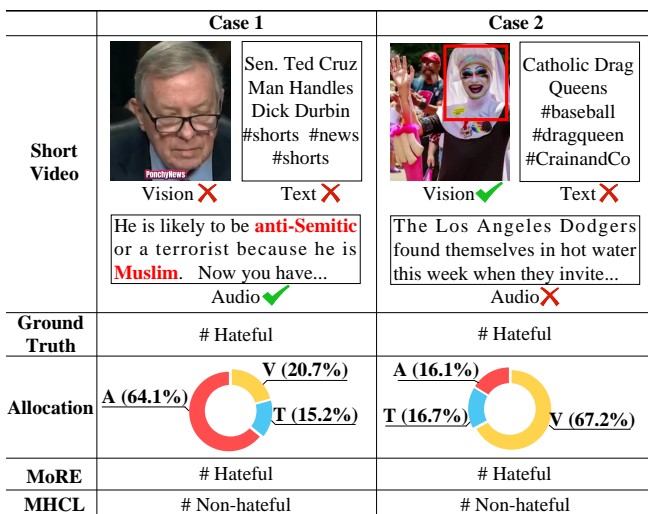

Fig. 6: Case study of the MoRE's explainability on dynamically assigning weights to modality experts for each video instance. A: Audio expert, T: Textual expert, V: Visual expert.

The first case illustrated on the left in Fig. 6 involves a short video where hateful content, specifically "anti-Semitic" and "Muslim", is presented solely in the audio modality. Our MoRE successfully captures the hateful evidence by prioritizing the audio expert, assigning it the highest weight (64.1%). The second case is more challenging, as neither the text nor audio contains hateful content. However, some frames in video show a group of men dressed as Catholic nuns mocking Christianity, which constitutes the hateful element. In this instance, MoRE effectively allocates the highest contribution (67.2%) to the visual expert, resulting in a correct identification of the hateful content. In contrast, the baseline model MHCL, which treats each modality equally, fails to detect the hateful content in these cases, leading to incorrect prediction. Additional case study with more short videos is provided in Appendix H.6.

## 5 Conclusion

In this work, we propose a novel MoRE framework to address SVHD. This multimodal framework leverages features from all modalities to enhance the precision of SVHD. A multimodal joint video retriever is developed to identify the most relevant instances for the target video. Multimodal experts gain contextual knowledge from these retrieved hateful and non-hateful instances, enhancing their ability to adapt to the dynamic evolution of hateful content. Additionally, a sample-sensitive integration network within MoRE adaptively adjusts the contributions of each expert based on different samples, further improving performance in SVHD. Furthermore, an end-to-end training paradigm is introduced to enhance the practical applicability of MoRE in real-world large-scale SVHD applications. Our extensive experiments conducted on three real-world datasets demonstrate the effectiveness of the proposed MoRE for SVHD. In the future, MoRE has the potential to become an important tool for many short video platforms like TikTok and YouTube Shorts, contributing to trustworthy AI, especially in the context of short video recommendation platforms.

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

## A  Additional Literature Review

### A.1  Multimodal Retrieval

Multimodal retrieval aims to retrieve the most relevant instances by leveraging information across different modalities, such as text, vision, and audio. Previous studies have primarily focused on text-image retrieval, with the objective of retrieving images that correspond to a given text query or text that corresponds to a given image query [22, 25, 34, 50, 55]. These earlier studies typically relied on models that did not employ pre-training, such as Convolutional Neural Networks (CNNs) [22] and Faster R-CNN [55], to extract representations from both image and text data. The introduction of powerful vision-language pre-trained models [20, 29, 37, 43, 52] has enabled researchers to develop methods that jointly encode text and image representations for more accurate retrieval. These models have demonstrated significant improvements in the quality of text-image retrieval tasks. With the growing popularity of short video content, video retrieval has become an increasingly important area of study. Many studies in video retrieval have focused on text-to-video retrieval, where a text query is used to retrieve relevant video content from large video collections [12, 17, 18, 38]. These approaches leverage pre-trained models to generate a common embedding space, facilitating the alignment of video and text representations. Despite advancements in text-to-video retrieval, limited research addresses video-to-video retrieval, where the goal is to find the most relevant video content given a video query. In this work, we propose a novel joint multimodal video retriever that integrates audio, textual, and visual modalities to enable comprehensive and precise video-to-video retrieval.

### A.2  Mixture of Experts

The Mixture of Experts (MoE) was first proposed by Jacob et al. [30] as a method to combine multiple experts, each trained on different subsets of data, into a single powerful model. Eigen et al. [21] extended the MoE concept to neural networks by incorporating a layer consisting of expert networks and a trainable gating mechanism. This gating mechanism assigns weights to the experts on a per-example basis, enabling MoE to produce a weighted combination of the experts' outputs. Recently, MoE has been extensively studied as a technique to enhance the model's capacity in terms of parameter size without incurring additional computational cost, particularly in the fields of natural language processing [23, 35, 58] and computer vision [26, 42, 49, 56, 59]. Switch Transformer [23] developed a sparse MoE architecture that improves sample efficiency in training by minimizing communication and computational overhead, making it effective for natural language processing tasks. In the multimodal learning domain, LIMoE [49] presented a sparse MoE model that allows for the simultaneous processing of both image and text using a contrastive loss during training. Much of the current work primarily focuses on using the sparsity of MoE to augment model parameters, overlooking one of the key strengths of MoE: the ability to dynamically adjust outputs based on the input data through expert routing. In contrast, our work first time introduces MoE into the task of video-based hate detection by designing contextual knowledge-augmented multimodal experts to tackle different modalities of the short video. Furthermore, a sample-sensitive integration network is proposed to identify the specific contributions of each modality expert's features to hate detection in each video.

## B  Feature Extraction

For the short video $S_i$, we start by extracting its initial information from each modality. Specifically, we isolate the audio component from the video, resulting in the audio representation $s_i^a$. Additionally, we uniformly sample $m$ key frames from the video, which contribute to the visual content information denoted as $s_i^v = \{\mathbf{v}_i^1, \mathbf{v}_i^2, \ldots, \mathbf{v}_i^m\}$. The textual information $s_i^t$ incorporates the title and description of the short video $S_i$.

To ensure alignment with prior research [13, 63] for fair comparison, we utilize the pre-trained BERT [16] and ViT [19] as textual and visual feature extractors. This allows us to derive the text features $\mathbf{x}_i^t \in \mathbb{R}^{n \times d_t}$ and visual features $\mathbf{x}_i^v \in \mathbb{R}^{m \times d_v}$, where $n$ is the number of word tokens, while $d_t$ and $d_v$ denote the dimensions of the textual and visual embeddings, respectively. Specifically, the visual embedding for each key frame is derived from the classification token in the last hidden states of the Vision Transformer (ViT), which serves as the global representation of the frame. For audio feature extraction, we compute the Mel Frequency Cepstral Coefficients (MFCC), resulting in audio features $\mathbf{x}_i^a \in \mathbb{R}^{l \times d_a}$, where $l$ denotes the number of audio frames, and $d_a$ represents the number of MFCC coefficients extracted from each audio frame.

## C  Complexity & Efficiency

In this section, we conducted detailed complexity and efficiency analyses of our proposed MoRE framework, offering insights into its practical applicability.

### C.1  Computational Complexity Analysis

We performed the computational complexity analysis of each main component within the proposed MoRE, which includes the complexity analysis of the joint multimodal video retriever, the contextual knowledge-augmented multimodal experts, and the sample-sensitive integration network.

- **Complexity of joint multimodal video retriever.** The time consumption during retrieval is primarily associated with calculating the cosine similarity between the query vector and the vectors of the stored samples. Let $d$ denote the dimension of the query vector and $m$ represent the total number of video samples in the memory bank $\mathcal{B}$. The time complexity for retrieving each video instance is $O(md)$. It is important to note that the retrieval process is independent of the overall framework and is conducted only once, effectively functioning as a data preprocessing step. Therefore, the overall time complexity of the joint multimodal video retriever can be considered negligible in the context of the entire framework.

- **Complexity of contextual knowledge-augmented multimodal experts.** The time consumption for contextual knowledge-augmented multimodal experts primarily stems from the Bipolar Hateful Attention Network (BHAN), which utilizes two attention mechanisms: $\text{Att}_{\text{Hat}}$ for hateful instances and $\text{Att}_{\text{Non}}$ for non-hateful instances. The attention mechanism itself involves operations over query, key, and value matrices, where

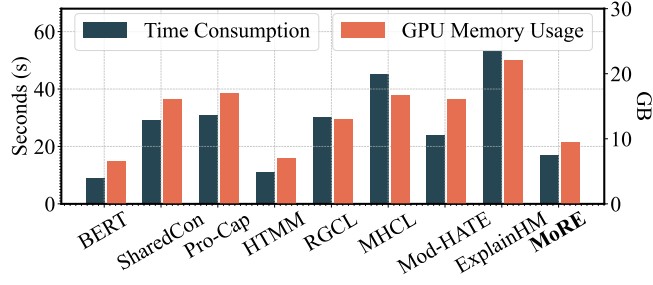

**Fig. 7: Runtime consumption and GPU memory usage of several baseline models and MoRE on the HateMM dataset.**

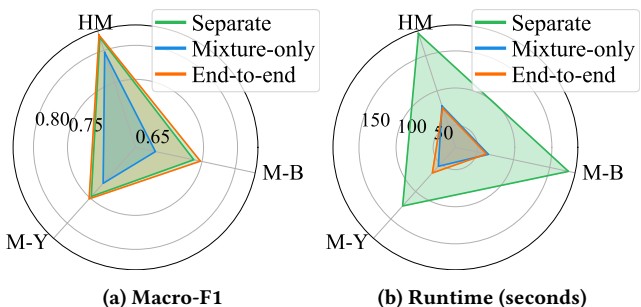

(a) Macro-F1      (b) Runtime (seconds)

**Fig. 8: Comparison of three training strategies on macro-F1 and runtime consumption across three datasets. HM: HateMM, M-Y: MHClip-Y, M-B: MHClip-B.**

the complexity is $O(d \cdot n^2)$ per attention head, with $d$ denoting the feature dimension and $n$ being sequence length. Since BHAN processes two attention mechanisms (hateful and non-hateful) for each modality, the overall time complexity for all three modality experts is $O(6 \cdot d \cdot n^2)$.

- **Complexity of sample-sensitive integration network.** The time complexity for the sample-sensitive integration network primarily stems from the modal-mixture soft router, which dynamically generates weights for modality fusion. The router is implemented as a two-layer MLP that computes the weights based on the input feature representations from each modality. The time complexity of this process for all three modalities is $O(3 \cdot d^2)$, where $d$ represents the feature dimension.

The overall complexity of MoRE stems from the matrix and vector operations, and the calculation of these operations can be accelerated by GPUs or TPUs. This design makes MoRE particularly well-suited for real-world deployments, especially for SVHD on social media platforms and applications.

## C.2 Efficiency Analysis

An empirical evaluation is conducted to assess the efficiency of the proposed framework, MoRE, in comparison to several competitive baseline models. This evaluation focuses on two key metrics: runtime consumption per epoch and GPU memory usage during the training phase. All experiments are performed on a single NVIDIA RTX 4090 GPU with a batch size of 64, using the HateMM dataset. The results are presented in Fig. 7.

As shown in the figures, BERT and HTMM demonstrate the lowest resource consumption in terms of both runtime and GPU memory, which can be attributed to their simpler model architectures. However, they struggle to effectively detect hate in short videos. In contrast, MHCL delivers relatively strong performance in SVHD but incurs substantial computational costs due to the complexity of its LSTM networks used as feature encoders. ExplainHM incurs the highest resource overhead, primarily due to the invocation of LVLMs for conducting a multimodal debate, the use of an additional LLM for judgment, and the fine-tuning of a T5 model for prediction. Although MoRE is not the most resource-efficient model, its impressive performance gains—achieving an average improvement of 6.91% in macro-F1 score across all three short video datasets—justify the additional computational overhead.

## D Training Strategy Comparison

In the previous section 3.4, we introduced the more efficient **end-to-end training paradigm**, which optimizes the entire MoRE framework for SVHD in a cohesive manner. In this section, we further demonstrate the effectiveness of this training paradigm by comparing it against two alternative strategies: (1) **Separate training**, which follows the prior MoE-based works [6, 70, 73]. Here, each expert network is first trained independently, and then a joint optimization strategy is applied to integrate the expert networks with the overall framework. (2) **Mixture-only training**, a more straightforward approach that directly optimizes the entire framework without any separate training for individual modality experts. In this approach, all components are optimized together in a single stage, bypassing any expert-specific fine-tuning.

To comprehensively assess the effectiveness of these training strategies, we conduct experiments on three datasets, evaluating them from two key perspectives: model performance with macro-F1 and training efficiency with runtime consumption. The results, presented in Fig. 8, lead to the following conclusions:

- **Performance evaluation**. As shown in Fig. 8a, the mixture-only training strategy experiences significant degradation in prediction performance. This issue likely arises from the lack of independent optimization for each expert network, which prevents the full utilization of their specialized capabilities. In contrast, both the separate training and the proposed end-to-end training emphasize fully training each expert network, enhancing their ability to detect hateful content specific to each modality, which significantly improves final predictions.
- **Efficiency evaluation**. Fig. 8b highlights a clear drawback of separate training: a substantial increase in runtime due to the computational cost of training each expert network independently, resulting in more than double the runtime compared to other methods. In contrast, both the mixture-only training and the end-to-end training approaches require only one stage of joint training, which substantially reduces runtime.

Compared to the computationally expensive separate training and the simplistic mixture-only training, our proposed end-to-end paradigm achieves competitive performance similar to separate training while significantly reducing computational costs to levels comparable with mixture-only training. By jointly optimizing the expert networks and the overall framework, it dynamically shifts the focus from expert-specific training in the early stages

**Algorithm 1** Training of MoRE for short video hate detection.

**Input:** The short video dataset $\mathcal{S} = \{S_1, \cdots, S_N\}$; The memory bank $\mathcal{B}$.

**Output:** Predicted category $\hat{y}$ (Hate or Non-hate).

1: **for each** instance $S_i$ in $\mathcal{S}$ **do**
2:     /* *Joint Multimodal Video Retriever* */
3:     Utilize all three modal queries $\mathbf{r}_i^a$, $\mathbf{r}_i^t$ and $\mathbf{r}_i^v$ of video $S_i$ to calculate the cosine similarities with samples in the memory bank $\mathcal{B}$ (Eq.(2)).
4:     Add the weighted similarities to obtain the final similarity scores, retrieving both hateful samples $S_i^r = \{S_i^{r,j}\}_{j=1}^K$ and non-hateful samples $\bar{S}_i^r = \{\bar{S}_i^{r,j}\}_{j=1}^L$ (Eq.(1)).
5:     /* *Contextual Knowledge-Augmented Multimodal Experts* */
6:     Feed the features $\mathbf{x}_i^m$, $m \in \{a, v, t\}$ from all modalities of $S_i$ into FFN to obtain the vanilla expert $\mathbf{E}_i^m$ (Eq.(3) - (4)).
7:     Encode the features of retrieved hateful instances $S_i^r$ and non-hateful instances $\bar{S}_i^r$ into FFN to generate the contextual knowledge $\mathbf{E}_i^{m,r}$ and $\bar{\mathbf{E}}_i^{m,r}$ (Eq.(3) - (4)).
8:     Equip the contextual knowledge to the vanilla expert $\mathbf{E}_i^m$ through BHAN to obtain the final representation of the modality expert $\mathbf{E}_i^{m,c}$ (Eq.(5) - (7)).
9:     /* *Sample-Sensitive Integration Network* */
10:    Average the modality features of $S_i$ and align them into a common feature space: $\Psi_a(\tilde{\mathbf{x}}_i^a)$, $\Psi_t(\tilde{\mathbf{x}}_i^t)$, $\Psi_v(\tilde{\mathbf{x}}_i^v)$. Input the aligned features into the soft router to assign weights for each modality expert $\mathbf{E}_i^{m,c}$ (Eq.(8) - (9)).
11:    Fuse the modality experts with the weights to obtain the final representation $\mathbf{E}_i$ of instance $S_i$ for prediction (Eq.(10)).
12:    /* *End-to-End Training* */
13:    Feed the $\mathbf{E}_i$ into the MLP based predictor to obtain the output $\hat{y}_i$. Input the representation of each modality expert $\mathbf{E}_i^{m,c}$ to the predictor to obtain the output $\hat{y}_i^m$.
14:    Jointly training the experts and the overall framework with BCE loss $L_{exp}$ and $L_{ovl}$ (Eq.(11) - (13)).
15: **end for**

to optimizing the entire network in the late stages. This training paradigm achieves a balance between prediction performance and computational efficiency, making it a compelling choice for practical deployment. The complete training algorithm for MoRE is provided in Algorithm 1.

# E Proof of Effectiveness of Contextual Knowledge-Augmented Multimodal Experts

The previous sections have demonstrated the motivation and process behind leveraging retrieved hateful and non-hateful instances to enhance the multimodal experts' ability to detect hate in short videos. In this section, the effectiveness of the contextual knowledge-augmented multimodal experts within the proposed MoRE framework is evaluated from an information-theoretic perspective. Specifically, for any given short video instance $S_i$, we aim to show that incorporating external knowledge from retrieved hateful and non-hateful instances $\mathbf{I}_i^R$ into the modality experts improves the prediction of the label $y_i$. Let $\mathbf{E}_i^a$, $\mathbf{E}_i^v$, and $\mathbf{E}_i^t$ represent the audio, textual, and visual modality experts of the instance $S_i$, respectively.

Without loss of generality, let $y_i$ denote the category of $S_i$. The following proposition is then presented.

PROPOSITION E.1. *Let the mutual information $I(X; Y)$ measure the amount of information transmitted between variables $X$ and $Y$. Then we have:*

$$I(y_i; \mathbf{E}_i^a, \mathbf{E}_i^v, \mathbf{E}_i^t, \mathbf{I}_i^R) \geq I(y_i; \mathbf{E}_i^a, \mathbf{E}_i^v, \mathbf{E}_i^t) \qquad (14)$$

**PROOF OF PROPOSITION E.1.** According to the definition of mutual information, we have:

$$I(y_i; \mathbf{E}_i^a, \mathbf{E}_i^v, \mathbf{E}_i^t, \mathbf{I}_i^R) = \mathbb{E}\left[\log \frac{\mathbb{P}(y_i, \mathbf{E}_i^a, \mathbf{E}_i^v, \mathbf{E}_i^t, \mathbf{I}_i^R)}{\mathbb{P}(y_i)\mathbb{P}(\mathbf{E}_i^a, \mathbf{E}_i^v, \mathbf{E}_i^t, \mathbf{I}_i^R)}\right]$$

$$= \mathbb{E}\left[\log \frac{\mathbb{P}(y_i, \mathbf{E}_i^a, \mathbf{E}_i^v, \mathbf{E}_i^t)\mathbb{P}(\mathbf{I}_i^R|y_i, \mathbf{E}_i^a, \mathbf{E}_i^v, \mathbf{E}_i^t)}{\mathbb{P}(y_i)\mathbb{P}(\mathbf{E}_i^a, \mathbf{E}_i^v, \mathbf{E}_i^t)\mathbb{P}(\mathbf{I}_i^R|\mathbf{E}_i^a, \mathbf{E}_i^v, \mathbf{E}_i^t)}\right] \qquad (15)$$

$$= \mathbb{E}\left[\log \frac{\mathbb{P}(y_i, \mathbf{E}_i^a, \mathbf{E}_i^v, \mathbf{E}_i^t)}{\mathbb{P}(y_i)\mathbb{P}(\mathbf{E}_i^a, \mathbf{E}_i^v, \mathbf{E}_i^t)}\right] + \mathbb{E}\left[\log \frac{\mathbb{P}(\mathbf{I}_i^R|y_i, \mathbf{E}_i^a, \mathbf{E}_i^v, \mathbf{E}_i^t)}{\mathbb{P}(\mathbf{I}_i^R|\mathbf{E}_i^a, \mathbf{E}_i^v, \mathbf{E}_i^t)}\right]$$

$$= I(y_i; \mathbf{E}_i^a, \mathbf{E}_i^v, \mathbf{E}_i^t) + I(y_i; \mathbf{I}_i^R|\mathbf{E}_i^a, \mathbf{E}_i^v, \mathbf{E}_i^t)$$

Since the conditional mutual information $I(y_i; \mathbf{I}_i^R|\mathbf{E}_i^a, \mathbf{E}_i^v, \mathbf{E}_i^t) \geq 0$, we can conclude that:

$$I(y_i; \mathbf{E}_i^a, \mathbf{E}_i^v, \mathbf{E}_i^t, \mathbf{I}_i^R) \geq I(y_i; \mathbf{E}_i^a, \mathbf{E}_i^v, \mathbf{E}_i^t)$$

Thus, the proof of the proposition is complete. □

Proposition E.1 reveals that the contextual knowledge learned from retrieved short video instances in the memory bank $\mathcal{B}$ encompasses more meaningful information compared to solely relying on the visual, audio, and textual modal information within a single instance. This finding underscores the effectiveness of our design in leveraging contextual knowledge to enhance the capabilities of multimodal experts in detecting hate in short videos, demonstrating that the integration of retrieved instances significantly contributes to the improvement of model's performance in SVHD.

# F Proof of Effectiveness of Sample-Sensitive Integration Network

In this section, we present the theoretical proof to demonstrate the superiority of our Sample-Sensitive Integration Network (SSIN) in detecting hateful content in short videos compared to traditional vanilla modality fusion strategies, which treat each modality equally during the fusion process. To align with the problem definition of SVHD provided in the main paper, we consider the three modalities scenarios, where $\mathbf{E}_i^a$, $\mathbf{E}_i^t$, $\mathbf{E}_i^v$ denote the audio, textual and visual modality experts for the short video $S_i$, respectively.

*Definition F.1 (Prediction Error).* Let $\mathcal{F}(\cdot)$ denote an arbitrary modality fusion strategy, $\mathcal{P}(\cdot)$ represent the predictor which receives the fused modality representations and generates the predicted category $\hat{y}_i$ of the short video $S_i$. Moreover, let $y_i$ be the ground truth of whether the video $S_i$ is hateful or non-hateful. The prediction error for $S_i$ is defined as the difference between the predicted output and the ground truth:

$$\delta(\mathcal{F}(\mathbf{E}_i^a, \mathbf{E}_i^t, \mathbf{E}_i^v)) = |\mathcal{P}(\mathcal{F}(\mathbf{E}_i^a, \mathbf{E}_i^t, \mathbf{E}_i^v)) - y_i|. \qquad (16)$$

As the ground truth $y_i$ is discrete, the value of $\delta(\mathcal{F}(\mathbf{E}_i^a, \mathbf{E}_i^t, \mathbf{E}_i^v))$ has only two cases:

$$\delta(\mathcal{F}(\mathbf{E}_i^a, \mathbf{E}_i^t, \mathbf{E}_i^v)) = \begin{cases} 0, & \text{if } y_i = \hat{y}_i, \\ 1, & \text{if } y_i \neq \hat{y}_i. \end{cases} \quad (17)$$

Following the prior works [39, 68], we concisely redefine the redundant, unique, and synergistic modality interaction scenarios:

*Definition F.2 (Redundancy).* In the modality redundancy scenario, all modalities contains the important information for accurate prediction and they contain redundant information.

*Definition F.3 (Uniqueness).* In the uniqueness scenario, only one or two modalities provides critical information for accurate prediction. In this case, equally fuse all three modalities may lead to error prediction while prioritizing one or two modalities can achieve the best result.

*Definition F.4 (Synergy).* In the synergy scenario, each modality provides distinct but complementary information that, when effectively combined, can achieve optimal predictions. In this case, correctly assigning weights to each modality lead to the best prediction.

PROPOSITION F.5. *Our proposed modality fusion strategy SSIN($\cdot$), which dynamically adjusts the contributions of audio, textual, and visual modalities for different video samples, achieves lower or equal prediction error compared to the Vanilla Fusion Strategy VFS($\cdot$) across all three modality interaction scenarios:*

$$\delta(SSIN(\mathbf{E}_i^a, \mathbf{E}_i^t, \mathbf{E}_i^v)) \leq \delta(VFS(\mathbf{E}_i^a, \mathbf{E}_i^t, \mathbf{E}_i^v)), \quad \forall S_i \in \mathcal{S}. \quad (18)$$

PROOF OF PROPOSITION F.5. To prove this proposition, we analyze the prediction errors of both SSIN($\cdot$) and VFS($\cdot$) across the three defined modality interaction scenarios: redundancy, uniqueness, and synergy. We begin by proposing some assumptions:

(1) *Modality Experts Performance*: Each modality expert $\mathbf{E}_i^m$ for modality $m \in \{a, t, v\}$ provides an estimate of the ground truth label $y_i$ with some probability of error $P_{\text{error},i}^m = \text{Pr}(\hat{y}_i^m \neq y_i)$.

(2) *Independence of Errors*: The errors made by different modality experts are statistically independent. This assumption simplifies the analysis and is reasonable when modalities are sufficiently different.

(3) *SSIN Weighting Mechanism*: SSIN assigns weights $w_i^m$ to each modality $m$ for sample $S_i$, based on the estimated reliability or informativeness of that modality for that sample.

We aim to demonstrate that for each $S_i$, the error probability with SSIN is less than or equal to that with VFS:

$$P_{\text{error},i}^{\text{SSIN}} \leq P_{\text{error},i}^{\text{VFS}}. \quad (19)$$

**Redundancy scenario.** In this scenario, all modalities are equally informative and have similar error probabilities:

$$P_{\text{error},i}^a = P_{\text{error},i}^t = P_{\text{error},i}^v = P_{\text{error}}. \quad (20)$$

*VFS*: VFS assigns equal weights to all modalities. The combined error probability is:

$$P_{\text{error},i}^{\text{VFS}} = P_{\text{error}} \left( 1 - (1 - P_{\text{error}})^2 \right). \quad (21)$$

*SSIN*: SSIN recognizes that all modalities are equally informative and assigns similar weights:

$$w_i^a = w_i^t = w_i^v = \frac{1}{3}. \quad (22)$$

The combined error probability is the same as VFS:

$$P_{\text{error},i}^{\text{SSIN}} = P_{\text{error},i}^{\text{VFS}}. \quad (23)$$

**Uniqueness scenario.** In this scenario, only a subset of modalities is informative. Without loss of generality, assume that modality $t$ is informative, while $a$ and $v$ are non-informative with error probabilities close to 0.5 (random guessing). *VFS*: VFS assigns equal weights, so the non-informative modalities dilute the informative signal. The combined error probability is higher due to the influence of noisy modalities. *SSIN*: SSIN assigns a higher weight to the informative modality $t$ based on its lower error probability:

$$w_i^t > w_i^a, \quad w_i^t > w_i^v. \quad (24)$$

By reducing the influence of noisy modalities, SSIN achieves a lower combined error probability:

$$P_{\text{error},i}^{\text{SSIN}} < P_{\text{error},i}^{\text{VFS}}. \quad (25)$$

**Synergy scenario.** In this scenario, each modality provides complementary information necessary for accurate prediction. *VFS*: Equally weighting modalities may not effectively capture the complementary strengths, potentially leading to suboptimal integration. *SSIN*: SSIN dynamically adjusts weights to optimize the fusion of complementary information. By assigning weights proportional to the informativeness of each modality, SSIN enhances the combined prediction capability:

$$P_{\text{error},i}^{\text{SSIN}} \leq P_{\text{error},i}^{\text{VFS}}. \quad (26)$$

**Formal error probability calculation.** Let us formalize the error probabilities under the assumption of independent modality errors. *VFS Error Probability*: For VFS, the combined prediction is based on a majority vote or average of the modality predictions. The error probability is:

$$P_{\text{error},i}^{\text{VFS}} = \sum_{\text{odd } k} \left( \sum_{m \in \{a,t,v\}} P_{\text{error},i}^m \right)^k \left( \sum_{m \in \{a,t,v\}} (1 - P_{\text{error},i}^m) \right)^{3-k}. \quad (27)$$

*SSIN Error Probability*: For SSIN, the combined prediction weights each modality according to its estimated reliability. The error probability is given by:

$$P_{\text{error},i}^{\text{SSIN}} = \text{Pr} \left( \sum_{m \in \{a,t,v\}} w_i^m \delta_i^m > \frac{1}{2} \right), \quad (28)$$

where $\delta_i^m$ is the individual modality prediction error (1 if incorrect, 0 if correct). Since SSIN assigns higher weights to more reliable modalities (lower $P_{\text{error},i}^m$), it reduces the overall error probability compared to VFS. **Conclusion** In all scenarios, SSIN either matches or outperforms VFS in terms of prediction error:

$$\delta(\text{SSIN}(\mathbf{E}_i^a, \mathbf{E}_i^t, \mathbf{E}_i^v)) \leq \delta(\text{VFS}(\mathbf{E}_i^a, \mathbf{E}_i^t, \mathbf{E}_i^v)), \quad \forall S_i \in \mathcal{S}. \quad (29)$$

Therefore, SSIN achieves lower or equal prediction error compared to VFS across all samples. □

By formally defining the error probabilities and demonstrating how SSIN adjusts modality contributions to minimize prediction errors, we have shown that SSIN is theoretically more effective than VFS in detecting hateful content in short videos.

**Table 6: Characteristics of three short video datasets.**

| Dataset Characteristic | HateMM | MHClip-Y | MHClip-B |
|---|---|---|---|
| **Total Videos** | 1,083 | 1,000 | 1,000 |
| **Hateful Videos** | 431 | 82 | 128 |
| **Offensive Videos** | N/A | 256 | 194 |
| **Non-Hateful Videos** | 652 | 662 | 678 |
| **Avg. Duration (s)** | 150.0 | 33.8 | 31.8 |
| **Languages** | English | English | Chinese |
| **Platforms** | BitChute | YouTube | Bilibili |

## G  Detailed Experimental Settings

### G.1  Datasets

We conduct comprehensive experiments to evaluate the performance of the proposed MoRE framework compared to baseline models on three real-world short video datasets: HateMM [13], MultiHateClip-YouTube (MHClip-Y), and MultiHateClip-Bilibili (MHClip-B) [63]. The characteristics of these datasets are outlined in terms of the total number of videos, the counts of hateful and offensive videos, non-hateful videos, average video duration, languages, and the platforms from which the videos were sourced, as shown in Table 6.

- **HateMM**: This dataset is a hateful video detection dataset, collected from *BitChute*, an alternative video-sharing platform with minimal content moderation. The English-language videos were manually annotated by trained annotators. Each entry contains the full video, and hate/non-hate label, with additional annotations including frame spans indicating hateful content and targeted communities.
- **MHClip-Y, MHClip-B**: These two datasets are benchmark datasets designed for hateful video detection on *YouTube* and *Bilibili*, respectively. Each entry in these two datasets includes the video, its title, transcript, and detailed annotations. The annotations provide rich information, including the video's classification (hateful, offensive, or non-hateful), specific hateful/offensive segments with timestamps, the target victim group (e.g., Woman, Man, LGBTQ+), and the contributing modalities (audio, textual, and visual).

Notably, we present the binary classification experimental results in the main paper by merging the "offensive" and "hateful" categories into a single "hateful" class. Additionally, multi-category classification experimental results (i.e., distinguishing between hateful, offensive, and non-hateful) are provided in Appendix H.2.

### G.2  Baselines

To validate the efficacy of MoRE, we compare our framework with competitive baseline models, which can be classified into three distinct groups: (1) *Unimodal hate detection methods*; (2) *Multimodal hate detection methods*; and *(3) Large Vision-Language Model (LVLM)-based methods*. Below, we provide detailed descriptions of each baseline.

(1) *Unimodal hate detection methods*:

- **BERT** [16]: Given the efficacy of BERT in hate speech detection [47], we employ BERT as a competitive unimodal baseline. The text data, including the video title, description, and audio transcription, is passed through BERT to extract features (i.e., the [CLS] token) represented in a 768-dimensional space. These features are subsequently fed into two fully connected (FC) layers to yield the final prediction results.
- **ViViT** [3]: The Video Vision Transformer is the video version of ViT [19], which is effective in video understanding and classification [54, 74]. We utilize ViViT to extract a 768-dimensional feature vector from 32 sampled frames for each video. The features are then input into two FC layers to generate the final output.
- **MFCC**: MFCC plays a pivotal role in audio signal processing and has been widely used in audio classification [5, 62]. For each video, we generate a 128-dimensional MFCC vector, which is then processed through two fully connected (FC) layers to obtain the final prediction results.
- **SharedCon** [1]: SharedCon designs a clustering-based contrastive learning approach that leverages the shared semantics among the data for implicit hate speech detection.

(2) *Multimodal hate detection methods*:

- **Pro-Cap** [8]: Pro-Cap utilizes prompting techniques to guide pre-trained vision-language models in generating image captions associated with hateful content. It subsequently combines these generated captions with textual information to enhance the detection of hateful memes.
- **HTMM** [13]: HTMM extracts features from transcripts, video frames, and audio frames. These features are then concatenated and input into an MLP-based classifier to detect hateful content in short videos.
- **RGCL** [46]: RGCL constructs a retrieval-based hateful memes detection framework which learns hatefulness-aware vision and language joint representations via an auxiliary contrastive objective and the dynamically retrieved examples.
- **MHCL** [63]: MHCL analyzes the significance of each modality in the detection of hateful content within videos. It then leverages the audio, textual, and visual features with LSTM-based feature encoders to perform hateful video detection.
- **Mod-HATE** [10]: Mod-HATE develops a suite of LoRA modules and employs few-shot learning to train a module composer that assigns weights to the modules based on their relevance. Subsequently, the weighted composition these LoRA modules generates the final prediction results.
- **ExplainHM** [40]: ExplainHM leverages a multimodal debate between LVLMs to generate opposing rationales from harmless and harmful perspectives. These rationales are judged by a LLM and are distilled to fine-tune a T5 model for the final prediction, ensuring both accuracy and explainability in harmful meme detection.

(3) *LVLM-based methods*:

- **MiniCPM-V** [72]: MiniCPM-V is a series of end-to-end VLLMs designed for vision-language understanding. These models accept text, images, and videos as inputs, generating high-quality text outputs. In this study, we adopt the latest and most advanced model in the MiniCPM-V series, MiniCPM-V 2.6, as our competitive baseline.

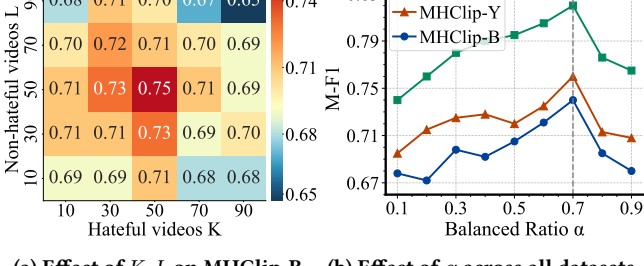

(a) Effect of $K$, $L$ on MHClip-B.  (b) Effect of $\alpha$ across all datasets.

Fig. 9: Additional sensitivity analysis of $K$, $L$, and $\alpha$. The metric used in (a) is M-F1.

Table 7: Example of prompt for hateful detection applied in LVLM-based methods.

---

**Prompt**: Now your task is to determine whether a short video is hateful or non-hateful based on its title, description, audio transcription and raw video content. If the video is hateful, output 1; otherwise, output 0.
**Video Title**: { Title }
**Video Description**: { Description }
**Audio Transcription**: { Transcription }
**Raw Video Content**: { Raw video content (in MP4 format) }
Now give your prediction (no need analysis, return 0 or 1 only).

---

- **LLaVA-OV** [36]: LLaVA-OneVision (LLaVA-OV) is the newest family of open MLLMs in the LLaVA series, which achieves new state-of-the-art performance across single-image, multi-image, and video benchmarks.
- **Qwen2-VL** [64]: Qwen2-VL is the latest version of the vision language models in the Qwen model families. Qwen2-VL has the abilities of complex reasoning and decision making and achieves state-of-the-art performance on visual understanding benchmarks.

Notably, for LVLM-based methods, we provide the text and raw video content along with a specifically designed prompt to guide the output generation. An example of the prompt is presented in Table 7.

## G.3  Implementation Details

In this section, we provide detailed implementation specifications for our proposed MoRE along with a comprehensive overview of the experimental setup.

- **Data processing.** We uniformly extract 16 key frames from each short video across all datasets to ensure consistent visual representation. To extract audio features, we employ the open-source library Librosa to compute the MFCC. For audio transcription, we employ two versions of the pre-trained Whisper [53] automatic speech recognition model, each separately fine-tuned for Chinese and English audio. To generate descriptions of the video content, we employ the pre-trained BLIP2 model, specifically the opt-2.7b version, to caption the extracted key frames. Additionally, we apply a chotomous image segmentation model

IS-Net [51] finetuned in background removal task to separate the background from the subjects in the key frames.
- **Details of memory bank construction.** In this work, the memory bank $\mathcal{B}$ is composed of short videos from the training and validation sets, thereby preventing data leakage during model testing. However, in real-world applications, the memory bank must be continuously updated to reflect temporal changes, ensuring that the model can adapt to the rapidly evolving nature of hateful content.
- **Training configuration.** During the retrieval, the default weight for each modality is set to equal. For text, we set the maximum sequence length to 512 for all datasets. For key frames, we resize the images into 224 × 224. The number of retrieved hateful videos $K$ and non-hateful videos $L$ are selected from the set {10, 20, 30, 40, 50}, respectively. And the bipolar attention balancing ratio $\alpha$ is chosen from the range [0, 1]. The positive constant $\delta$ in end-to-end training is set to 0.2. We utilize the AdamW [41] optimizer with a learning rate of $5 \times 10^{-4}$ and a weight decay of $5 \times 10^{-5}$ for model parameters optimization. We set the random seed to 2024. For statistical testing, where each model is run five times, we use random seeds ranging from 2024 to 2028 and report the mean value as experimental results. For baseline models, we strictly adhere to the settings specified in their original papers.
- **Implementation environment.** All experiments are conducted on a system equipped with an Intel(R) Core(TM) i9-14900KF processor, an NVIDIA GeForce RTX 4090 GPU with 24 GB of VRAM, and 128 GB of system RAM.

## H  Additional Experiments

### H.1  Hyper-Parameter Analysis

We also conduct the sensitivity analysis of $K$ and $L$ on the MHClip-B dataset, along with an evaluation of another hyper-parameter, $\alpha$ (the bipolar attention balance ratio), across all three datasets.

- **Number of retrieved videos $K$ and $L$.** We analyze the sensitivity of the parameters $K$ and $L$ on the MHClip-B dataset, with the results presented in Fig. 9(a). The findings corroborate the conclusions drawn in the main paper: performance initially improves as the number of retrieved videos of both types increases. However, including a large number of instances ultimately results in a decline in performance, primarily due to the noise introduced by irrelevant videos. For the MHClip-B dataset, we select $K = 50$ and $L = 50$ to achieve optimal performance.
- **Bipolar attention balanced ratio $\alpha$.** We evaluate the sensitivity of the parameter $\alpha$ with the BHAN and the results are shown in Fig. 9(b). From the results, we observe that MoRE performs best across all datasets when $\alpha = 0.7$, reflecting a heightened focus on hateful attention. By assigning a higher weight to hateful attention, MoRE incorporates more contextual knowledge from hateful instances. This aligns with the distribution of the dataset labels, where hateful videos are relatively scarce, thus requiring greater attention to capture the necessary contextual knowledge from retrieved instances.

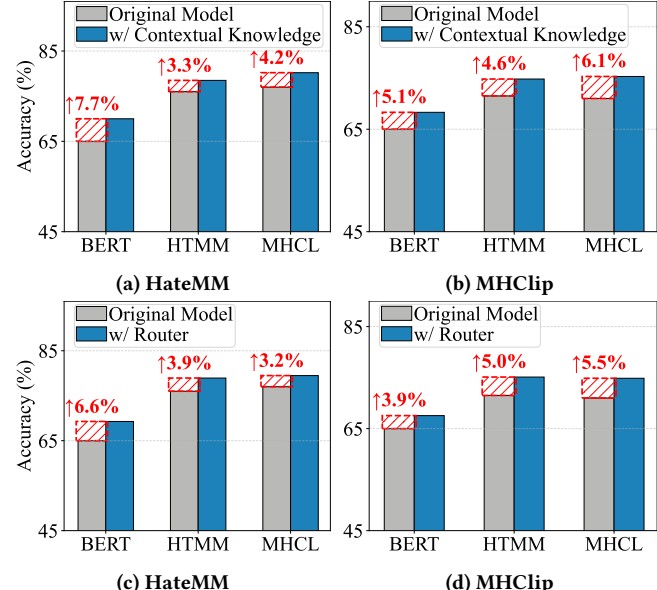

**Fig. 10: Performance of several baseline models: BERT, HTMM and MHCL with and without main components of MoRE on the HateMM and MHClip-Y datasets.**

**Table 8: Additional experimental results for the three-class classification task on the MHClip-Y and MHClip-B datasets. The best results are in red bold, while the second results are in black bold. Higher values of ACC, M-F1, and M-P indicate better performance.**

| Method | MHClip-Y | | | MHClip-B | | |
|---|---|---|---|---|---|---|
| | ACC | M-F1 | M-P | ACC | M-F1 | M-P |
| BERT | 0.6101 | 0.4752 | 0.4771 | 0.5245 | 0.4614 | 0.4617 |
| AHSC | 0.5980 | 0.3615 | 0.3742 | 0.5133 | 0.3184 | 0.3345 |
| Pro-Cap | 0.6461 | 0.4942 | 0.4768 | 0.6377 | 0.4876 | 0.4994 |
| HTMM | 0.6533 | 0.4979 | 0.4814 | 0.6677 | 0.4767 | 0.4959 |
| MHCL | 0.6733 | 0.5108 | 0.5299 | 0.6544 | 0.4852 | 0.5096 |
| MiniCPM-V | 0.6578 | 0.5110 | 0.5284 | 0.6746 | 0.5033 | 0.5352 |
| LLaVA-OV | 0.6711 | 0.5095 | 0.5323 | 0.6876 | 0.5101 | 0.5668 |
| Qwen2-VL | **0.7067** | **0.5134** | **0.5814** | **0.6907** | **0.5129** | **0.5774** |
| **MoRE** | **0.7133** | **0.5324** | **0.5923** | **0.7001** | **0.5216** | **0.5959** |
| $p$-val. | $5.11e^{-2}$ | $1.26e^{-2}$ | $3.24e^{-2}$ | $6.43e^{-2}$ | $7.13e^{-2}$ | $1.87e^{-2}$ |

## H.2 Multi-Category Classification

To further validate the superior performance of our proposed MoRE, we also conduct multi-category classification (i.e., hateful, non-hateful, or offensive) experiments on the MHClip-Y and MHClip-B datasets and the results are presented in Table 8. From the table, we have the following observations:

**(O1): All the models meet the performance drop.** In the multi-category classification task, the imbalanced distribution of classes—approximately 8% hateful, 26% offensive, and 66% non-hateful—poses significant challenges for model training and inferring. Furthermore, the minimal content differences between hateful and offensive videos exacerbate the classification difficulty, as the models struggle to effectively differentiate between these two closely related categories. Overall, these factors contribute to the observed decline in performance across all models, including our proposed MoRE.

**(O2): LVLM-based methods exhibit strong performance.** Despite the significant class imbalance in the dataset, the performance of LVLM-based models remains robust. This can be attributed to their zero-shot inferring paradigm, which allows these models to leverage their extensive knowledge acquired during pre-training stage without requiring additional fine-tuning.

**(O3): MoRE outperforms all competitive baselines.** Our proposed framework MoRE leverages contextual knowledge from videos across all three categories, enabling the expert to capture subtle distinctions among them. Additionally, MoRE dynamically assigns weights to each modality expert based on sample characteristics, allowing precise detection of offensive and hateful content in short videos. Consequently, MoRE effectively addresses the multi-category hate detection in short videos, demonstrating its superiority framework design.

## H.3 Model Scalability

In this section, we analyze the scalability of our proposed MoRE by integrating its core components into various baseline models to demonstrate their broad applicability and effectiveness in enhancing performance on detecting hate in short videos.

*H.3.1 Scalability of Contextual Knowledge-Augmented Multimodal Experts.* First, we integrate the contextual knowledge from the retrieved hateful and non-hateful videos into baseline models BERT, MHCL, and HTMM, and perform the experiments on the HateMM and MHClip-Y datasets. Specifically, we add the BHAN into these baseline models and the results are presented in Fig. 10(a) and Fig. 10(b). From the results, we observe that incorporating this extra knowledge significantly improves the performance of these models on both datasets. The integrated contextual information provides up-to-date insights into evolving hateful content and enables the models to better distinguish between hateful and non-hateful instances. This highlights the significance of utilizing additional knowledge to improve the model's capability to detect evolving hateful content.

*H.3.2 Scalability of Sample-Sensitive Integration Network.* Next, we implement the sample-sensitive integration network to replace the fusion strategies employed in these baseline models and conduct experiments on the HateMM and MHClip-Y datasets. Specifically, we add the router network to these baseline models and the results are provided in Fig. 10(c) and Fig. 10(d). Our results indicate that dynamically assigning weights to different modalities at the sample-level significantly enhances precision in SVHD. This improvement stems from the fact that hate can manifest in various modalities across different short video samples, highlighting the necessity for a more flexible fusion strategy to achieve precise detection.

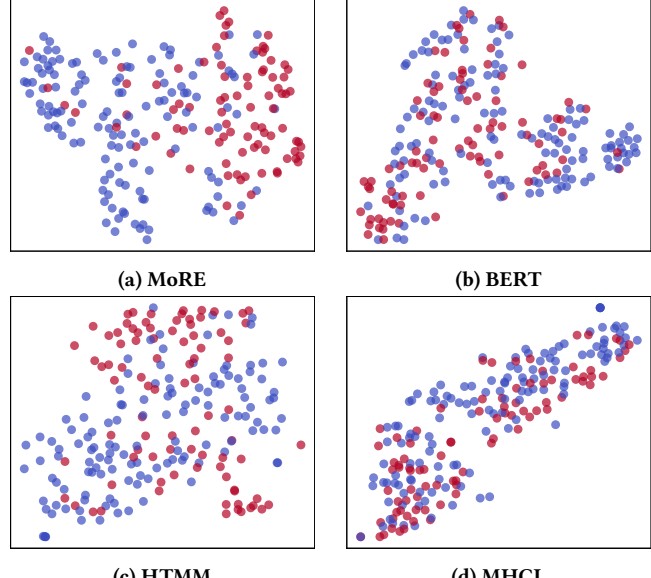

**(a) MoRE**      **(b) BERT**

**(c) HTMM**      **(d) MHCL**

Fig. 11: t-SNE visualization of MoRE and several baseline models: BERT, HTMM, and MHCL on the HateMM dataset. Red points indicate the hateful samples while blue points represent the non-hateful samples.

## H.4 Visualization

We employ t-SNE [61] to visualize the embedding space of two categories: hateful and non-hateful, and the results are presented in Fig. 11. The embedding used in this study is the output from the last layer of the classifier in each model. We observe that while both HTMM and MHCL learn distinguishable features, their embeddings remain somewhat entangled between the two categories. BERT, on the other hand, fails to separate the categories effectively, showing the weakest performance. In contrast, our MoRE produces more discriminative representations, with clearer boundaries between instances of different labels. This result underscores MoRE's ability to capture the nuanced distinctions between hateful and non-hateful content in short videos by leveraging contextual knowledge from retrieved instances and dynamically assigning weights to each modality expert at the sample level, resulting in more accurate predictions for detecting hateful videos.

## H.5 Retrieval Quality Presentation

In this section, we further evaluate the retrieval quality by randomly selecting more short video samples from the HateMM and MHClip-Y datasets. These samples encompass both their retrieved hateful and non-hateful instances, offering a comprehensive view of the effectiveness of our proposed joint multimodal video retriever. Designed to capture similarities across audio, textual, and visual modalities, the retriever ensures that the retrieved instances closely match the target videos in all relevant aspects. The results are presented in Table 9 and Table 10.

## H.6 Case Study: Model Explainability

We randomly select more short videos to demonstrate the explainability of our MoRE in assigning varying contributions to different modality experts, depending on the presence of hateful content in each modality. The results are shown in Fig. 12.

## I Limitations and Future Works

In this section, we provide multiple ways which can further improve our work in detecting hate in short videos.

- **Enhancing the retriever with trainable components.** In this work, we adopt a training-free strategy in our proposed joint multimodal video retriever, which is resource-efficient for video-video retrieval. However, trainable components can be incorporated into the retriever, allowing for optimization via contrastive learning. This approach could potentially lead to better adaptation to retrieval tasks in SVHD and may improve the precision of search results.
- **Enriching the memory bank with more valuable instances.** In the future, the memory bank $\mathcal{B}$ could be expanded to include more valuable short video instances, which may enhance the quality of contextual knowledge provided to the modality experts and has the potential to improve performance in detecting hate in short videos.
- **Improving the explainability of the MoRE framework.** In this study, the explainability is provided by the router in the sample-sensitive integration network, limiting it to modality-level explainability in hate video detection. In the future, we plan to incorporate additional components into our proposed MoRE to offer more fine-grained explainability, allowing for deeper insights into how each modality contributes to the detection process.

## J Broader Impacts of Our Work

Our proposed MoRE has the potential to make a significant social impact by providing short video platforms with a more effective tool to identify and reduce harmful content. By targeting hate speech directed at marginalized groups based on race, gender, and other forms of discrimination, MoRE helps foster safer online environments for users. Moreover, MoRE's adaptive capability allows it to keep pace with evolving hate expressions, ensuring that the system remains effective as new forms of harmful content emerge. This adaptability offers a sustainable solution for platforms that need to continuously update their detection capabilities to meet the challenges posed by rapidly changing online behaviors. Additionally, MoRE introduces an efficient training paradigm that significantly reduces resource consumption, making it well-suited for deployment on short video platforms. This efficient strategy not only improves the scalability of the model in handling large-scale video data but also facilitates its real-world application, ensuring faster and more effective detection of hateful content in a high-throughput environment. Finally, MoRE provides a basic level of explainability, offering content moderators and viewers insights into why a video has been classified as hateful. This transparency fosters greater trust in short video hate detection systems and supports video moderators in their decision-making process during content moderation.

**Table 9: Presentation of the retrieval quality from the HateMM dataset. H: Hateful, N: Non-hateful. "V / A / T" refers to the cosine similarity scores between the target video and the retrieved videos across visual, audio, and textual modalities.**

| | Target: H | Top-1: H | Top-2: H | Top-1: N | Top-2: N |
|---|---|---|---|---|---|
| **Vision** |  |  |  |  |  |
| **Audio** | why did you put this racist black male, this nigger in the same prison as, you know, with a white male... | I'm sleeping And your stinky nigger smell in my brain Still remains, I is a white trans-neighbor... | Just as the slave master in that day used Tom, the house negro, to keep the field niggers in check... | All white people get shot and it's a federal case now? I claim black man, I didn't pay no mind.... | Arno Michalis knows better than most the mind of a far right-wing white racist in the world, He says... |
| **Text** | Racist nigger attacks and kills White; After Meeting With black male Pastor N Sup.. | Hello nigger my old friend; You've come to rob my house again you niggers softly crip | Nigger Barkley on Racist and Police N; Breaking News New Details on Officers... | We're way past black and white; I told you the truth and nothing but the truth... | racist skinhead organization in the world; He says he was one white supremacist ... |
| **V / A / T** | N/A | 0.70 / 0.93 / 0.88 | 0.66 / 0.85 / 0.90 | 0.68 / 0.90 / 0.89 | 0.66 / 0.86 / 0.91 |
| | **Target: N** | **Top-1: H** | **Top-2: H** | **Top-1: N** | **Top-2: N** |
| **Vision** |  |  |  |  |  |
| **Audio** | Jews were imbued with hatred of Christian Europeans. In Israel terrorizing Christians and Muslim Palestinians... | That is the right thing. Are you Jewish? So we know how the Jewish people discriminate against Christians... | Take me to church. I have to confess my sins to the Muslim. The number 9 is coming to my mind. 9 Christians... | Last year, he angered Jewish groups when he sang these words. When Christ did things to free the oppressed... | The Holocaust is big business because Germany so far has given the Jews, Jewish ships, I think, at least... |
| **Text** | The Jewish role in violence South Africa; terrorizing European Christians and Muslim Palestinians... | The godly thing is to kill me; how the Jewish people feel about Christians; yes Christians are idol worshlpers... | Can I Take You To church; we are all for Muslim, out to kill all you Christians, but don't forget, we are... | Rabbi Marvin Heyer, an activist for Jewish causes, who was outraged; I said, Christ did things ... | Then people don't realize who brought the slaves to America. They were Jewish ships; Wait a minute... |
| **V / A / T** | N/A | 0.61 / 0.90 / 0.91 | 0.65 / 0.89 / 0.86 | 0.72 / 0.87 / 0.86 | 0.70 / 0.80 / 0.77 |

**Table 10: Presentation of the retrieval quality from the MHClip-Y dataset. H: Hateful, N: Non-hateful. "V / A / T" refers to the cosine similarity scores between the target video and the retrieved videos across visual, audio, and textual modalities.**

|  | Target: H | Top-1: H | Top-2: H | Top-1: N | Top-2: N |
|---|---|---|---|---|---|
| Vision |  |  |  |  |  |
| Audio | Oh, my mom said that I'm a prostitute. I don't charge body for sex. I give man a way for free... | That is, I look like a prostitute and I am charging the richest man in the world for sex... | Top 10 Countries with most prostitutes Countries; Top10 Dominican Republic, Top9... | In Greek legend, Phryne, famous prostitute, the god have given her this body... | Ramchandra Khan sent a very beautiful, very attractive prostitute to Shri Haridas Thakur... |
| Text | I give a way for free and pay 530; Because I am a lady; Mom called me a prostitute... | I am a prostitute; I am a lady of the evening dropped pants and the prices... | Top 10 Countries With Most Prostitutes in the world; short sex education edits... | In Ancient Greek Prostitute tried to escape death; famous prostitute avoid showing... | Prostitute Sent to Defame; WATCH FULL VIDEO on (Secret to become spiritual)... |
| V / A / T | N/A | 0.75 / 0.93 / 0.90 | 0.69 / 0.77 / 0.74 | 0.81 / 0.89 / 0.87 | 0.71 / 0.67 / 0.75 |
|  | Target: H | Top-1: H | Top-2: H | Top-1: N | Top-2: N |
| Vision |  |  |  |  |  |
| Audio | Straight women hate men more than anybody. Because lesbians can't hate men that much because queers don't have to deal with them... | I discovered lesbian people and holy, holy hell. When I had that first, one of my first queer breakups, the physical pain hit me. It may... | The question is, who speaks to children about the lesbian? The mom and dad family, the person who has given her life to raising... | If you're outing, No, I overheard her telling a friend that she is lesbian, the queer, non-binary and I am obligated by the school to tell you... | I am lesbian and I love that this person is just fully playing a game right behind you. She's deaf. She's deaf? My bad. You do your boo-boo.... |
| Text | Chosen family podcast; lesbians can not hate man; if you go on line and you meet queers; #gay #lgbt... | lesbians share #queer break-up stories ashgavs Makingemi chosenfamilypod #love #dating #heartbreak #lgbt... | Disney Forces Lesbian Couple Into Lightyear #Shorts #Lightyear #Disney #LGBT #AndrewKlavan #DailyWire | As queer Parents at School #shorts #lgbt #gay; Follow Me, I have something to tell you about your child... | lesbian comedian accidentally roasts deaf person #lgbt #pride #comedy #jokes #stand up; I love this person... |
| V / A / T | N/A | 0.84 / 0.91 / 0.94 | 0.71 / 0.79 / 0.84 | 0.78 / 0.88 / 0.92 | 0.81 / 0.76 / 0.81 |

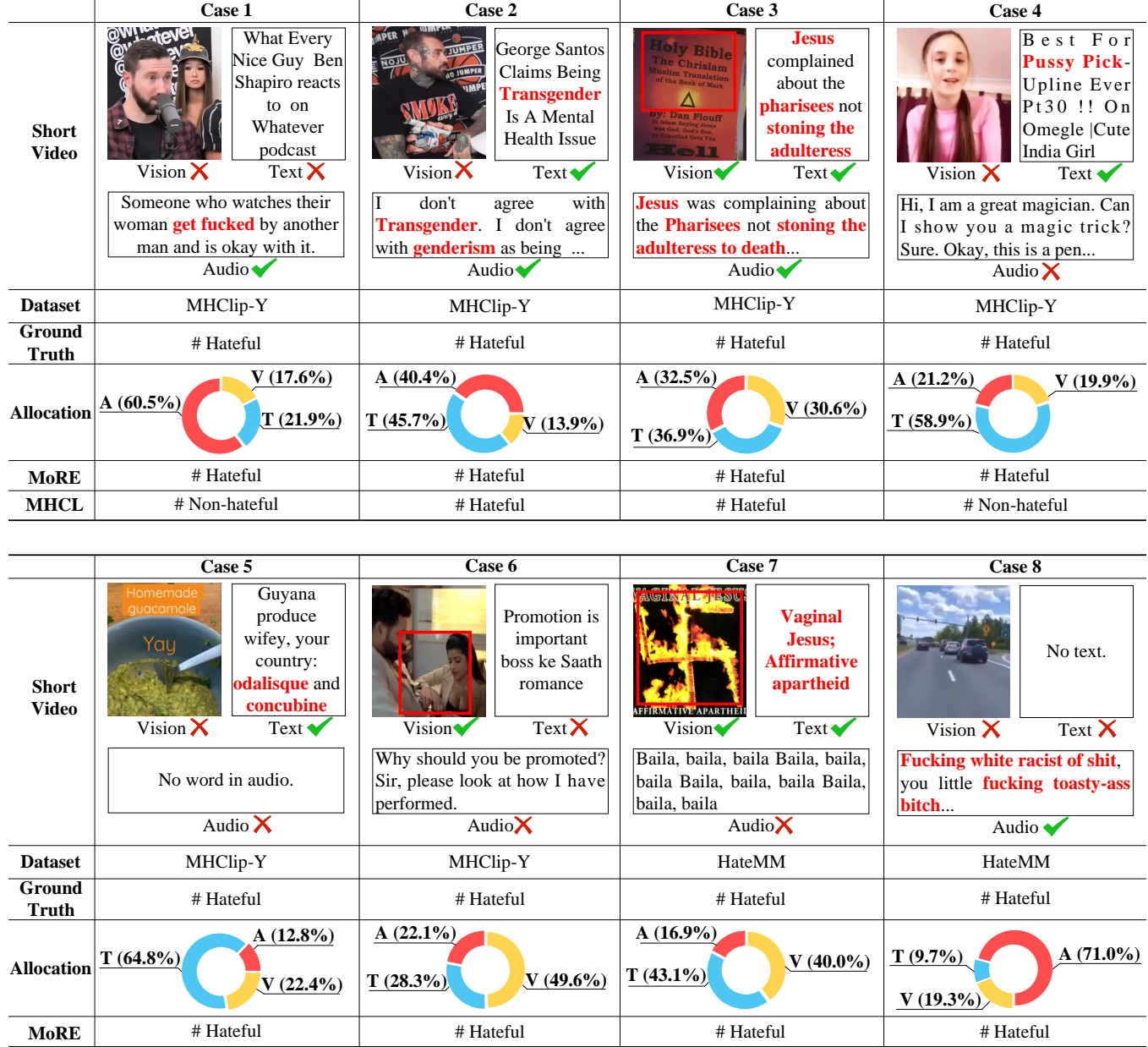

Fig. 12: Case study of the MoRE's explainability on dynamically assigning weights to modality experts for each video instance. **A: Audio expert, T: Textual expert, V: Visual expert.**

