# OpenReview forum: "Biting Off More Than You Can Detect: Retrieval-Augmented Multimodal Experts for Short Video Hate Detection"
_ACM.org/TheWebConf/2025/Conference — WWW 2025 Poster_

### Official Review · Reviewer_snR7 · 2024-11-12

**Novelty:** 4
**Technical Quality:** 5

**Review:**

Summary:

This paper proposes a model for short video hate detection using multimodal inputs, along with an efficient end-to-end training approach. The model consists of three primary components: a Joint Multimodal Video Retriever, Contextual Knowledge-Augmented Multimodal Experts, and a Sample-Sensitive Integration Network. The Joint Multimodal Video Retriever generates representations for text, audio, and visual modalities, and retrieves relevant videos based on a weighted similarity score. The Contextual Knowledge-Augmented Multimodal Experts enhance each modality of the original video by incorporating both hateful and non-hateful instances, using the Bipolar Hateful Attention Network (BHAN) to provide enriched context to each expert. The Sample-Sensitive Integration Network dynamically adjusts the weight of each modality expert based on sample features, enabling a more accurate determination of whether the video contains hateful content. The end-to-end training approach jointly optimizes both the expert networks and the overall framework, significantly enhancing computational efficiency.

Strengths:

1.The model’s structure is comprehensive, including the Joint Multimodal Video Retriever, Contextual Knowledge-Augmented Multimodal Experts, and the Sample-Sensitive Integration Network. By leveraging contextual information from retrieved hateful and non-hateful instances, the multimodal experts adapt effectively to the dynamic evolution of hateful content. The Sample-Sensitive Integration Network adaptively adjusts each expert’s contribution, further enhancing detection accuracy.

2.The end-to-end training method not only improves computational efficiency but also allows the expert networks and overall framework to be jointly optimized.

3.Extensive experiments on the HateMM, MultiHateClip-Youtube (MHClip-Y), and MultiHateClip-Bilibili (MHClip-B) datasets demonstrate that this approach outperforms state-of-the-art models in terms of accuracy, macro-F1 score, macro precision, and macro recall. Cross-dataset training and evaluation validate the model’s generalizability.

4.Comprehensive ablation studies and correctness validation experiments demonstrate the model’s accuracy and effectiveness.

Weaknesses:

1.The model processes audio by converting it to text for encoding, which may overlook audio-specific features such as tone and pauses. Visual modality processing through keyframe extraction and averaging could miss frame-to-frame relationships and differences in frame importance. Using methods like fusion-in-decoder from text processing could yield better feature extraction.

2.While the end-to-end training method improves computational efficiency, the paper does not provide evidence that it has no negative impact on model performance, making it difficult to rule out any potential adverse effects compared to traditional methods.

3.Some gains reported in the ablation studies lack data representation and are only discussed qualitatively.
Certain baseline models, such as BERT, are relatively outdated; more recent models could enhance comparisons.

**Questions:**

1.Given the growing role of large language models (LLMs) in short video detection, do you think incorporating an LLM could further enhance your model’s performance?

2.Are the weights for the three-modal similarity in Equation 1 fixed, or do they vary dynamically by sample? How are the weights in the retriever determined?

3.How is the balance coefficient  between hateful and non-hateful attention determined in Equations 6 and 7?

4.Is there a fixed formula for the linear mapping function in Equation 8?

5.How is the small non-zero constant  in Equation 10 determined?

6.Why does the Modality-Mixture Soft Router (MSR) use a two-layer MLP instead of more layers? What impact would more or fewer MLP layers have on MSR performance?

**Reviewer Confidence:**

4: The reviewer is certain that the evaluation is correct and very familiar with the relevant literature

**Scope:**

4: The work is relevant to the Web and to the track, and is of broad interest to the community

---

### Official Review · Reviewer_wopR · 2024-11-29

**Novelty:** 6
**Technical Quality:** 6

**Review:**

The paper seems technically sound and well-structured. The proposed approach is innovative in integrating retrieval-augmented contextual knowledge with a sample-sensitive integration network for multimodal hate detection. The experiments are comprehensive, covering benchmarks, ablation studies, and sensitivity analyses.

Pros:
+ Addresses a significant real-world problem with societal impact.
+ Introduces novel techniques for retrieval augmentation and sample-sensitive integration.
+ Evaluation demonstrates superiority over state-of-the-art methods.

**Questions:**

No questions.

**Reviewer Confidence:**

1: The reviewer's evaluation is an educated guess

**Scope:**

4: The work is relevant to the Web and to the track, and is of broad interest to the community

---

### Official Review · Reviewer_BNda · 2024-12-02

**Novelty:** 5
**Technical Quality:** 5

**Review:**

The paper presents a well-structured and comprehensive approach to the problem of hate detection in short videos, specifically through the MoRE framework. The methodology is clear and the experiments are thorough. The work is original in its approach to integrating multimodal data (audio, text, and visual) for hate detection in short videos. The significance of this work lies in its potential impact on improving content moderation on social media platforms, which is a pressing issue in today's digital landscape. The findings could lead to better detection systems that adapt to evolving forms of hate speech.

**Strength:**

-- Innovative approach combining multiple modalities for hate detection.

-- Good results supported with necessary analysis (*).


**Weakness:**

-- The generalizability of the model across different platforms could be further explored.

-- Potential limitations regarding the dataset diversity and its impact on model performance.

-- * It would be great to see some error cases along with some justification where MoRE failed.

**Questions:**

**1.** How do you determine the value of "α" when computing the Hate and Non-Hate attention? What effects do different values of "α" (e.g., 0, 1, 0.5) have on the results?

**2.** Is there a mechanism to ensure that the model does not inadvertently reinforce biases present in the training data? If so, how MoRE can address it?

**Reviewer Confidence:**

3: The reviewer is confident but not certain that the evaluation is correct

**Scope:**

4: The work is relevant to the Web and to the track, and is of broad interest to the community

---

### Official Review · Reviewer_Bj6n · 2024-12-02

**Novelty:** 6
**Technical Quality:** 6

**Review:**

The paper introduces MoRE (Mixture of Retrieval-augmented Multimodal Experts), a novel framework designed to enhance Short Video Hate Detection (SVHD) by addressing critical challenges such as the evolving nature of hateful content, multimodal complexity (audio, text, vision), and the varying contributions of each modality to hate detection. MoRE employs specialized multimodal experts equipped with contextual knowledge retrieved via a joint video retriever, enabling better adaptability to emerging hate expressions. Additionally, a sample-sensitive integration network dynamically adjusts modality weights based on their relevance to hate detection for each video. The framework adopts an end-to-end training strategy, jointly optimizing both the experts and the overall architecture, significantly improving computational efficiency. Extensive experiments on three datasets demonstrate MoRE’s superiority, achieving a 6.91% average improvement in macro-F1 score over state-of-the-art baselines.

Strengths

●	Utilizes a comprehensive approach that leverages multimodal data (text, audio, vision) and dynamically adjusts weights based on modality relevance.

●	Introduces a joint multimodal video retriever and a sample-sensitive integration network to address evolving and context-specific hate content.

●	Demonstrates superior performance with an average macro-F1 improvement of 6.91% over baseline models across three datasets, showcasing practical utility and strong empirical validation.

●	Evaluates adaptability to unseen data, highlighting the model's generalizability and robust transfer capabilities between platforms.

●	Provides explainability by offering insights into modality contributions and the model’s decisions through weight allocation analysis.

●	Improves computational efficiency with an end-to-end training paradigm, jointly optimizing all components instead of optimizing each component independently.

Weaknesses:

●	Considering that the model’s performance (including the creation of the memory bank and training process) heavily relies on the quality and relevance of the retrieved videos, the authors should consider implementing measures to mitigate the impact of noise in retrieval. One approach is to include noise filtering techniques and another approach is to incoporate multi-stage retrieval systems, where at the initial stage identifies a broad set of instances, followed by a fine-grained filtering stage to ensure only the most relevant and high-quality instances are retained.

●	Evaluation is conducted on only three datasets; incorporating broader benchmarks could enhance the assessment of generalizability.

MultiHateClip: A novel multilingual dataset created using hate lexicons and human annotation, designed to detect hateful videos on platforms like YouTube and Bilibili, with content in both English and Chinese.

HSDVD Dataset: From Rana, Aneri, and Sonali Jha's study "Emotion-Based Hate Speech Detection Using Multimodal Learning" (arXiv, 2022). This dataset includes multimodal hate speech with an emphasis on emotional cues.

YouTube Religious Hate Speech Dataset: Curated by Noman Ashraf et al., focusing on detecting religious hate speech and extremism in YouTube videos (YouTube Based Religious Hate Speech and Extremism Detection Dataset with Machine Learning Baselines, 2022).

Flickr30K Entities and Hateful Memes (Facebook AI): Although not short videos, it could assess the model's multimodal (text combined with visual) hate detection capabilities.

●	The authors could enhance the audio expert by improving audio preprocessing through advanced noise reduction and normalization techniques to ensure clean and high-quality inputs. Audio Expert could be seperately optimized with audio-focused hate speech detection benchmark, such as AudioHateXplain , improving its ability to detect subtle hate signals in tone, pitch, or speech patterns.

**Questions:**

Questions

●	How does the joint retriever handle noisy or irrelevant instances, and are there measures to filter such data during retrieval?

●	What strategies are being considered to enhance the performance of the audio modality expert?

●	Given the retrieval and integration components, what are the runtime requirements for MoRE, and is it viable for real-time hate detection?

●	Does including instances from the target dataset in the memory bank during generalizability testing affect the true evaluation of the model's ability to generalize to unseen domains? Would the results differ significantly if the memory bank were constructed from unrelated datasets?

●	How much is computational efficiency improved by using the end-to-end training strategy compared to independently optimizing each component? Are there measurable trade-offs in terms of convergence speed or model performance?

●	Can the framework be fine-tuned incrementally as new data becomes available, or does it require full retraining?

●	Does joint optimization ever lead to conflicts between the objectives of individual expert networks?

**Reviewer Confidence:**

2: The reviewer is willing to defend the evaluation, but it is likely that the reviewer did not understand parts of the paper

**Scope:**

4: The work is relevant to the Web and to the track, and is of broad interest to the community

---

### Official Review · Reviewer_cq5F · 2024-12-03

**Novelty:** 5
**Technical Quality:** 4

**Review:**

Quality: The paper is well-executed with strong empirical support, but lacks details on scalability and computational complexity.

Clarity: The paper is well-structured, clearly identifying key challenges in SVHD and systematically describing the proposed solutions. Figures and diagrams support understanding.

Originality: The MoRE framework introduces originality in its retrieval-based contextual augmentation and sample-sensitive fusion of modalities. The use of Bipolar Hateful Attention Network (BHAN) to incorporate both hateful and non-hateful instances for context is a novel aspect.

Significance: The work is significant in addressing the evolving and multimodal nature of hateful content on popular short video platforms. The proposed end-to-end framework effectively adapts to dynamic hateful expressions.

Pros:

The sample-sensitive integration network improves accuracy by dynamically adjusting modality contributions.

The joint multimodal video retriever effectively adapts experts to evolving hate content.

Extensive experiments, comparisons, and case studies demonstrate consistent performance gains.

The unified end-to-end approach enhances scalability and practicality.

Achieves notable gains (6.91% increase in Macro-F1) over state-of-the-art baselines.

Cons:

Retrieval may introduce computational overhead, limiting real-time applicability.

Missing detailed discussion on deployment challenges and complexity.

Textual features are often prioritized, reducing effectiveness for non-verbal cues.

Some methodology parts are mathematically dense and hard to follow.

More discussion is needed on cultural sensitivity, false positives, and biases.

**Questions:**

How does the joint multimodal video retriever scale in real-time scenarios or large-scale datasets?

Could you provide more insights into the computational complexity of the proposed framework, especially concerning the retrieval and training phases?

How can the framework ensure balanced detection of non-verbal cues, such as subtle visual or audio signals, without over-prioritizing text?

Are there any visualizations or interpretability methods that could help to understand why a specific modality received higher weight?

**Reviewer Confidence:**

2: The reviewer is willing to defend the evaluation, but it is likely that the reviewer did not understand parts of the paper

**Scope:**

4: The work is relevant to the Web and to the track, and is of broad interest to the community